# Quality assessment and umbrella review of systematic reviews about dance for people with Parkinson's disease

Camila Pinto[1,2], Rafaela Simon Myra[3], Alexandre Severo do Pinho[2], Francisca Pereira[2], Guido Orgs[4]*, Aline Souza Pagnussat[1,2,3]

**1** Department of Physical Therapy, Universidade Federal de Ciências da Saúde de Porto Alegre, Porto Alegre, Rio Grande do Sul, Brazil, **2** Health Sciences Graduate Program, Universidade Federal de Ciências da Saúde de Porto Alegre, Porto Alegre, Rio Grande do Sul, Brazil, **3** Rehabilitation Sciences Graduate Program, Universidade Federal de Ciências da Saúde de Porto Alegre, Porto Alegre, Rio Grande do Sul, Brazil, **4** Institute of Cognitive Neuroscience, University College London, London, United Kingdom

* g.orgs@gold.ac.uk

**Data Availability Statement:** All relevant data are within the manuscript and its Supporting Information files.

## Abstract

### Objective(s)

To determine (1) the quality of systematic reviews about dance-based intervention in individuals with Parkinson's disease (PD) and (2) standard evidence for dance-based intervention efficacy based on the categories of The International Classification of Functioning, Disability, and Health (ICF) from the World Health Organization's (WHO).

### Methods

The data source included MEDLINE, PUBMED, Embase, Scopus, CENTRAL (Cochrane Library), CINAHL, PEDro, SPORTDiscus, APA PsycNet (APA PsycINFO), LILACS, SciELO, and AMED. Pairs of independent reviewers screened titles, abstracts, and full texts of eligible studies by using the software Covidence. Criteria included: systematic review designs; individuals with PD; dance-based interventions aimed to change critical PD symptoms matched to IFC domains (body functions, activities, and participation). Independent reviewers extracted information regarding the characteristics of all systematic reviews included and appraised quality using A MeaSurement Tool to Assess Systematic Reviews (AMSTAR 2). Randomized controlled trials and their risk of bias were identified within each review and were used to perform an updated pairwise meta-analysis.

### Results

Of the 571 manuscripts screened, 55 reviews met the inclusion criteria. The overall confidence in the results of 38 reviews (69%) was rated as 'critically low,' nine (9%) as 'low,' one (2%) as 'moderate,' while seven of 55 reviews (13%) were rated as 'high'. Dance associated with pharmacological usual care is better than pharmacological usual care alone for essential components of ICF, such as motor symptoms severity (body function), depressive symptoms (body function), balance (body function and activity), and functional mobility (activity),

**Funding:** The author(s) received no specific funding for this work.

**Competing interests:** The authors have declared that no competing interests exist.

**Abbreviations:** PD, Parkinson's Disease; ICF, The International Classification of Functioning, Disability and Health; WHO, World Health Organization; AMSTAR, A MeaSurement Tool to Assess Systematic Reviews; PROSPERO, International Prospective Register of Systematic Reviews; PRIOR, Preferred Reporting Items for Overviews of Reviews; RCT, Randomized Clinical Trial; GRADE, Grading of Recommendations Assessment, Development and Evaluation; UPDRS III, Unified Parkinson's Disease Rating Scale, Motor Section; BDI, Beck's Depression Inventory; BBS, Berg Balance Scale; TUG, Timed Up and Go; 6MWT, Six Minute Walking Test; PDQ-39, Parkinson's Disease Questionnaire.

but not for gait distance (activity) and quality of life (participation). Dance is also superior to multimodal exercise to improve balance.

## Conclusions

Clinicians and people with PD can refer to this paper for a summary of high-quality reviews and the overall evidence supporting dance as an adjunct rehabilitation. This umbrella review not only underscores the therapeutic potential of dance but also reinforces the use of arts-based approaches into healthcare practices for people with neurological conditions.

## Introduction

According to the World Health Organization (WHO), Parkinson's disease (PD) is the second most common neurodegenerative movement disorder in the world (8.5 million in 2019), and prevalence has doubled in the past 25 years. The International Classification of Functioning, Disability and Health (ICF) by the World Health Organization (WHO) [1] is a classification of health categories to assess health well-being by looking at the individual patient in a more integrative way. The ICF expands on the biopsychosocial perspective of PD disability in terms of body function (e.g. balance), activities (e.g. functional mobility such as turning, walking, and sitting to standing), and participation (e.g. quality of life) by including personal and environmental factors [2–4] into the assessment. The negative influences of PD motor and non-motor symptoms on daily living tend to progress over time, so optimizing ICF domains is the overall goal of PD management [2, 3]. The most known treatment with pharmacological usual care (e.g. levodopa) does not cover all PD symptoms, and side effects, such as dyskinesias and anxiety, can become more severe with chronic administration [5, 6]. For this reason, evidence-based practices reinforce the use of nonpharmacological approaches, in particular, rehabilitation with active physical exercises [7, 8]. Exercise would have a potential to alleviate anxiety associated with levodopa treatment in individuals with PD. When people diagnosed with PD do not receive proper nonpharmacological treatment, the disease progresses more rapidly, the need for institutionalization increases and the costs can become very high for the patient and the family [9, 10].

Dance has emerged as both a popular and effective intervention to enhance the quality of life in PD. Dance provides health benefits similar to other forms of exercise or multimodal training [2, 11], yet as an art-based form of body and emotional expression it has additional social and cognitive benefits associated with the learning and remembering of movement sequences, improvisations, storytelling and collective movement guided by music. Dance serves multiple purposes including physical exercise, recreational programs, and rehabilitation. European [2] and American [12, 13] Guidelines for physiotherapy include dance as a therapeutic, multimodal, and community-based intervention recommended for people with PD. Multimodal training interventions encompass three or more combined modalities or components of physical exercises, such as balance, aerobic resistance, motor coordination, and flexibility, among others. The European Physiotherapy Guideline for PD (2014) [2] recommends dance for improving functional mobility, but not as an intervention to support quality of life or motor symptoms severity [2]. More recently, the American Guideline (2022) [12] provides strong recommendations for community-based exercise to improve functional mobility and quality of life, and reduce motor symptoms severity. It includes activities such as dance, yoga, Pilates, but does not provide any specific evidence that dance can improve PD

symptoms or quality of life. Future research should focus on a target modality due to the range of variability between dance and other community-based modalities [12].

The primary source of guidelines for clinical practice are systematic reviews and meta-analyses [14]. Over the past ten years, several systematic reviews [15–20] and meta-analyses [11, 21–35] have shown the efficacy of concomitant in-person dance-based interventions for individuals with PD and compared these to pharmacological usual care alone or other types of physical exercise. A meta-analysis published in 2020 showed a significant improvement in quality of life after dance-based intervention in individuals diagnosed with PD (SMD = −0.30; 5 RCTs) compared to pharmacological usual care [36]. Another meta-analysis published in 2021 showed benefits on motor severity (MD -6.91; 5 RCTs) and balance (MD 4.47; 3 RCTs) when comparing dance-based intervention to pharmacological usual care, but no difference in quality of life [25]. Overall, while there appears to be some consensus that dance is beneficial, it is unclear if dance is primarily beneficial to quality of life, or also improves specific PD symptoms, i.e. balance.

It is therefore important to help clinicians and patients to identify the best reviews, i.e. those that base their recommendations on a meta-analysis and are transparent with respect to their limitations, such as heterogeneity, low-quality clinical trials and risk of bias [37]. As a solution, AMSTAR (A MeaSurement Tool to Assess Systematic Reviews) was developed in 2007 [38] to evaluate the quality of systematic reviews with or without meta-analysis. This tool was upgraded in 2017 as AMSTAR 2 [39] and presents crucial domains that can critically affect the validity and conclusions of a review if the domains are not followed.

The contradictions of guidelines and reviews evaluating the efficacy of dance-based interventions in people with PD justify assessing the methodological quality of systematic reviews with the help of the AMSTAR 2 tool. As mentioned above, there are inconsistent findings between guidelines and systematic reviews regarding the efficacy of dance-based intervention. Thus, based on ICF domains, a question remains about how the variability of movement strategies used in dance would effectively improve body functions, activities, and participation in people with PD. Thus, it is timely and important to systematize low-bias randomized clinical trials included in these reviews and update these findings through a pairwise meta-analysis.

## Objectives

Firstly, we aim to appraise the overall confidence of systematic reviews with and without meta-analysis about dance-based intervention for people with PD by using AMSTAR 2 tool. Secondly, we aim to provide standard evidence of high-quality RCTs included in meta-analyses of systematic reviews selected. Thus, this study brings two questions:

a. To what extent can we rely on systematic reviews and meta-analysis recommendations to assess the efficacy of dance-based interventions' for people with PD?

b. Does dance-based intervention improve ICF domains (body function, activity, and participation) in people with PD when compared to pharmacological usual care alone or other active exercises?

## Methods

Our paper provides an umbrella review of existing reviews on the efficacy of dance-based interventions for PD). The protocol for this study has been registered in the International Prospective Register of Systematic Reviews (PROSPERO) under the registration CRD42023413814. This review was reported based on the Preferred Reporting Items for Overviews of Reviews (PRIOR) [40] and Cochrane Collaboration's recommendations for

conducting an overview of reviews (Chapter V) [41]. The PRIOR statement was developed in 2022 [40] based on aspects of Preferred Reporting Items for Systematic Review and Meta-Analysis Protocols (PRISMA-P) 2020 [42] but focused on particular challenges in reporting overviews.

## Eligibility criteria

According to the eligibility criteria (PICOT), any study designed as a systematic review (if the review authors had identified it as such) with or without meta-analysis aimed to include (P) individuals with PD at any stage of the disease severity or duration; (I) dance-based intervention associated or not with pharmacological usual care; (C) compared with pharmacological usual care alone, other active intervention, or minimal intervention (e.g. education); (O) critical outcomes for PD management that are related to IFC domains (body functions, activities, and participation), such as motor symptoms severity, depressive symptoms, balance, functional mobility, walk distance, and quality of life; (T) post-treatment follow-up at least for four weeks of intervention. Studies that were not fully published in peer-review journals or designed as an overview of systematic reviews were excluded. This review has no publication time restriction and no language restriction.

**Search methods.** We performed a comprehensive search strategy with support from a health sciences librarian. We started the search on March 2023 and consulted the databases MEDLINE, PUBMED, Embase, Scopus, CENTRAL (Cochrane Library), CINAHL, PEDro, SPORTDiscus, APA PsycNet (APA PsycINFO), LILACS, SciELO, and AMED. We finished the selection in May 2023 and updated the search in June 2024. We also checked reference lists of included articles, prospero registries, grey literature and performed a hand search of general and specific journals. Health professionals with backgrounds in dance and PD were consulted as experts to ensure we did not miss any additional reviews. We contacted authors by email if the full paper was not available online. The search terms comprised "parkinson", "dance", and "review", as well as their synonyms and MESH terms adapted for distinct databases. Although the search was based on PICOT, we did not include search terms of outcomes and time to guarantee high sensitivity. S1 Table details the complete search strategy.

**Data collection and selection.** Two independent reviewers (CP and RSM) performed the search and screened titles and abstracts according to the eligibility criteria. To facilitate this systematic approach, we used the Covidence Software. Then, two reviewers (CP and RSM) independently read full articles eligible, any disagreements were resolved through discussion with a third (ASP) reviewer when necessary.

**Data extraction and management.** Two authors performed the data extraction independently (CP and RSM or FP) and then cross-checked each other's work. We extracted data for all included reviews into a built data extraction form to allow a qualitative overview of all the available evidence. We only included reviews on dance interventions in the context of PD. If reviews covered forms of exercise beyond dance, then data on these interventions was not extracted.

We extracted information from all RCTs included in all reviews that performed pairwise or network meta-analyses. Title, author's names, and *doi* of RCTs were extracted and filled out into a worksheet. Two authors (CP and FP) extracted all the data outcomes from the original RCT individually (not from meta-analyses conducted in the reviews). We did not extract data from NRCT because RCTs in this field provide the necessary outcome data and a complete picture of the efficacy associated with dance intervention [39]. After completing data extraction, we managed to evaluate the RCTs that implemented random methods of treatment allocation and were in line with PICO eligibility criteria of this review. We extracted the

methodological quality of RCTs based on PEDro scale classified by the Physiotherapy Evidence Database Team (www.pedro.org.au). Then, the authors filled all outcomes into a report sheet based on ICF patient assessment components [2].

## Assessment of quality of included reviews

Two reviewers (CP and RSM) evaluated methodological quality independently with disagreements resolved by the third author (ASP). The quality of systematic reviews with or without meta-analysis was assessed using the AMSTAR 2 (A MeaSurement Tool to Assess Systematic Reviews) [39]. The tool contains a checklist of sixteen items (16 AMSTAR-2 items) that are rated as yes (positive result), partial yes, or no (negative result), with other response options for cases if the review did not conduct a meta-analysis. Importantly, for those reviews that included exercises other than dance, we considered only review results of dance-based interventions for items 8, 9, 11, 12, 13, 14, and 15 of AMSTAR 2. There are critical and non-critical domains in AMSTAR 2. If critical domains are not fully followed, the rate of overall confidence will drop. The study's overall confidence is categorized into four rates that range from high, moderate, low or critically low, calculated according to *https://amstar.ca/Amstar_Checklist. php*. With high overall confidence, for example, the review presents no or just one non-critical domain or weakness, providing an accurate and comprehensive summary of the results of the studies that address the question of interest. Reviewers learned how to use AMSTAR-2 by referred literature and carried out a training trial before starting the evaluation of articles included in this paper. We used a previous quality assessment paper already published to learn and train our evaluation and discuss disagreements with an expert in the field. This training trial took around one month. Finally, each reviewer (CP and RSM) independently assessed the included articles and compared the results. Discrepancies were handled in a consensus dialogue in a group and remaining disagreements were resolved by a third author (ASP).

## Data synthesis and criteria for meta-analyses

The summary characteristics of systematic reviews included were synthesized qualitatively. Moreover, we provided a quantitative synthesis by re-analyzing the efficacy of dance for people with PD. For this purpose, we compared a dance-based intervention in addition to pharmacological usual care with pharmacological usual care alone or in combination with other active exercises. Outcomes related to the International Classification of Functioning, such as motor symptom severity (body function), depressive symptoms (body function), balance (body function and activity), functional mobility (activity), gait distance (body function and activity), and quality of life (participation), were assessed. Motor symptom severity was evaluated using the Unified Parkinson's Disease Rating Scale, Motor Section (UPDRS III). Depressive symptoms were assessed using Beck's Depression Inventory (BDI). Balance was measured using the Berg Balance Scale (BBS). Functional mobility was evaluated with the Timed Up and Go (TUG) test. Gait distance was assessed using the Six Minute Walking Test (6MWT). Quality of life was evaluated using the Parkinson's Disease Questionnaire (PDQ-39). We performed a meta-analysis based on at least two similar studies with respect to interventions, comparators, and outcomes [41]. There are several randomized and non-randomized studies about our field of study; however, most of them confuse the results because of their high risk of bias. Thus, we excluded quasi or only partially randomized clinical trials, and we only included RCTs with scores of six or higher rated as 'good' or 'excellent' on the PEDro scale, ensuring a low risk of bias and thereby providing better evidence for guidelines in meta-analyses. The data extracted from RCTs included pre- and post-intervention assessments. All RCTs included in meta-analyses used the same test to assess outcomes of interest, thereby substantially reducing the

heterogeneity of our results. For quantitative synthesis, data were pooled using a random effect model. Regarding the continuous outcomes, if the unit of measurement was consistent across trials, the results were presented as the weighted mean difference with 95% confidence intervals (CIs). Statistical heterogeneity was assessed using I2 statistics and classified as follows: low heterogeneity (I2 below 25%); moderate heterogeneity (I2 between 25% and 50%); high heterogeneity (I2 above 50%). A P value ≤.05 was considered statistically significant. All analyses were conducted using the R statistical software (version 3.3.3, package metaphor version 2.0–0).

## Certainty of evidence

The overall certainty of evidence was assessed using Grading of Recommendations Assessment, Development and Evaluation (GRADE). This tool aims to score the certainty of the body of evidence for each outcome as "high" (++ ++), "moderate" (+++), "low" (++), or "very low (+). The assessment included judgments about risk of bias, imprecision, inconsistency, indirectness, and publication bias [43, 44]. An evidence profile and summary of findings tables for each population was created using GRADE's electronic tool GRADEpro GDT (*www.gradepro.org*).

## Results

PRISMA flow diagram is detailed in Fig 1. From 571 records identified from databases, registries, and other methods, 135 reports were assessed for full consideration according to

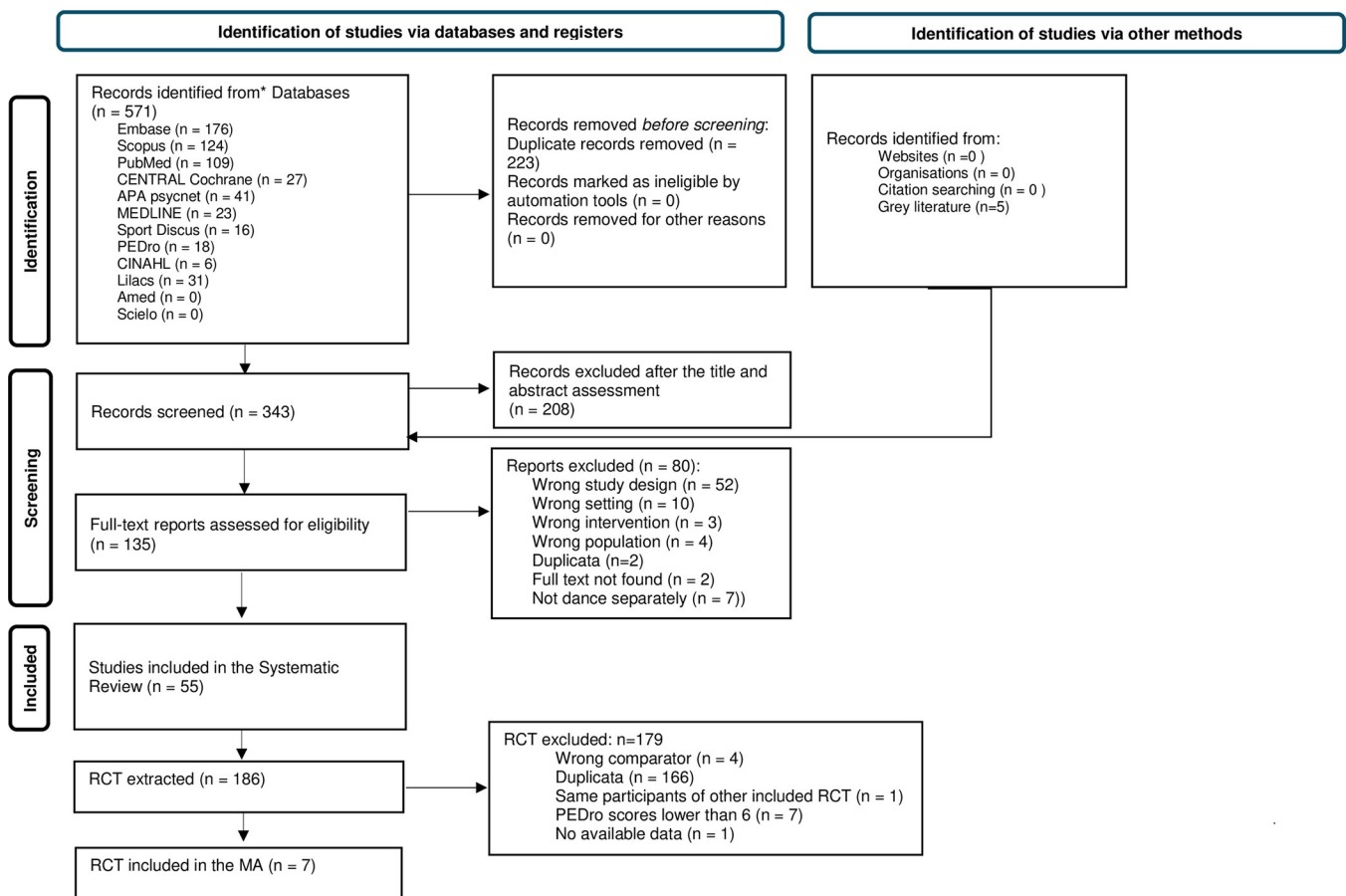

**Fig 1. PRISMA flow diagram.** Reference: Page MJ, McKenzie JE, Bossuyt PM, Boutron I, Hoffmann TC, Mulrow CD, et al. The PRISMA 2020 statement: an updated guideline for reporting systematic reviews. BMJ 2021;372:n71. doi: 10.1136/bmj.n71.

eligibility criteria and 55 reviews were included in the review (40 with meta-analysis and 15 systematic reviews without meta-analysis).

Tables 1, 2 presents the overall characteristics and results of 55 systematic reviews with and without meta-analysis included. In terms of patients, 53 systematic reviews investigated only the PD population, except for two reviews that included other types of neurodegenerative diseases [45] or physical illnesses [46]. Regarding intervention, 25 reviews investigated only dance as the experimental group, and 30 reviews covered other types of exercises [23, 26, 30, 35, 36, 47–71].

## Quality assessment of included reviews

Table 3 shows the rating overall confidence and each of the 16 items of AMSTAR 2 used to access 55 systematic reviews included. 38 reviews (69%) were rated as 'critically low' on quality (with more than one critical flaw with or without non-critical weaknesses), nine reviews (16%) were rated as 'low' (with only one critical flaw with or without non-critical weaknesses), one review (2%) was rated as 'moderate' (more than one non-critical weaknesses), and seven reviews (13%) were rated as 'high' (none or only one non-critical weakness).

From seven systematic reviews rated as 'high', only one was directly aimed at investigating dance-based intervention [25]. The others had a broader focus on physical exercise [36, 50, 70], therapeutic exercise [56], rehabilitation interventions [55], and mind-body exercises [53]. The systematic review rated as 'moderate' aimed to investigate the benefits of physical exercise [51]. The AMSTAR 2 critical domains and the percentage of total studies that did not fulfill the domain are summarized below:

- *Item 2*: not establishing a protocol before conducting the review (58%)

- *Item 4*: not appraising all relevant aspects of the literature search (94%)

- *Item 7*: not providing adequate justification for excluding individual studies (38%)

- *Item 9*: not reporting risk of bias from individual studies being included in the review (30%)

- \**Item 11*: Appropriateness of meta-analytical methods (25%)

- *Item 13*: not discussing the impact of risk of bias in the interpretation of the results (38%)

- \**Item 15*: not explaining in detail the presence and likely impact of publication bias (52%)

  \*Items with an asterisk represent the percentage of those 40 reviews with meta-analysis.

## Umbrella review

We identified 186 RCTs from included systematic reviews and meta-analyses. The exclusion of RCTs comprised:164 duplicate RCTs; four RCTs that compared different styles [72, 73] or sets [72] of dance; one RCT that associated dance with physiotherapy sessions [74]; one that evaluated the same participants as another RCT already included; and seven RCTs were rated lower than six on the PEDro scale [75–81]. The RCT of *Shanahan et al. (2017)* [82] with Irish dance was not included in the analysis due to unavailable data and no response after contacting authors by email. Thus, seven RCTs with 'good' or 'excellent' scores on the PEDro scale were considered for our meta-analysis. These scores can be found at Physiotherapy Evidence Database Team (www.pedro.org.au). These studies were from Italy, United Kingdom, Ireland, the US, Japan and the Republic of Korea. Five studies compared dance (Argentine tango dance [83], Irish dance [82], Qi dance [84], Tango dance [85], Sardinian folk dance [86], and mixed repertoire dance [87]) relative to pharmacological usual care. An exception is that the study of

**Table 1. Characteristics of included reviews with MA.**

| Authors | Year | Title | Dance Specific | Country | RCT (Dance-based) | Non-RCT (Dance-based) | Risk of Bias Evaluation of Clinical Trials | Participants | Intervention Group from Meta-Analysis (Dance-based) | Control Group from Meta-Analysis (vs. Dance-based) | Primary Outcome Measures Results from Meta-Analysis (Number of Studies) |
|---|---|---|---|---|---|---|---|---|---|---|---|
| Wei-Hsin Cheng, et al. | 2024 | The effect of dance on mental health and quality of life of people with Parkinson's disease: A systematic review and three-level meta-analysis | Yes | Australia | 12 studies | No | Yes (Cochrane Risk of Bias tool) | PD | Simple dance movements (including rumba and waltz), Sardinian folk dance, Argentine tango, Waltz/Foxtrot, Modern dance, Dance for PD program, Turo (Qi Dance), Irish set dance, Brazilian samba dance, Choreographed dance movement and improvisational movement | Exercise, Treadmill, Stretching, Physiotherapy Deutscher Karate, Tai Chi, No intervention, Usual care, Usual activities and routine, Wait-list. | Dance x passive control: ↑AS (2), ↑BDI (3), ↑HADS-anxiety (2), ↑HADS-depression (2), ↑CESD (1),↑GDS (1),↑SDS (1),↑Starkstein Apathy scale (1), ↑PDQ-39 (6), ↑BMLSS (1), ↑PDQL (1) Dance x active control: ↹ for all the scales cited above |
| Ernst M., et al. | 2024 | Physical exercise for people with Parkinson's disease: a systematic review and network meta-analysis (Cochrane Review) | No | Germany | 12 studies | No | Yes (Cochrane Collaboration's Tool for Assessing Risk of Bias 2) | PD | Tango, Tango or Waltz/Foxtrot, Sardinian folk dance, Irish set dancing, Partnered dance, Dance therapy, Music-based intervention, The Ronnie Gardiner Rhythm, and Music Method | Exercise, No intervention, Usual Care, Support group, Tai Chi, Tai Chi or no intervention | ↑ UPDRS III (5), ↑PDQ-39 (4) |
| Donida G., et al. | 2023 | Efectos de la danza sobre el equilibrio de personas con la enfermedad de Parkinson: una revisión sistemática con metaanálisis | Yes | Brazil | 3 studies | No | Yes (Cochrane Risk of Bias tool) | PD | Argentine Tango and Irish set dance | Usual care and daily activities and No intervention | ↑ Mini-BESTest (3) |
| Caroline Simpkins & Yang | 2023 | Do dance style and intervention duration matter in improving balance among people with Parkinson's disease? A Systematic review with Meta-analysis | Yes | USA | 12 studies | 2 studies | Yes (PEDro) | PD | Tango, Waltz/Foxtrot, Irish set dancing, Sardinian folk dance, Mixed styles (modern, aerobics, jazz, tango, ballet), Mixed ballroom, Turo (Qi Dance), Dance Therapy | Exercise, Physiotherapy, No intervention, Education, Support group, Treadmill, Stretching, Self-directed exercise, Physiotherapy + PD exercises | ↑ BBS (9), ↑ Mini-BESTest (4), ↑ FAB Scale (1) |

*(Continued)*

**Table 1.** (Continued)

| Authors | Year | Title | Dance Specific | Country | RCT (Dance-based) | Non-RCT (Dance-based) | Risk of Bias Evaluation of Clinical Trials | Participants | Intervention Group from Meta-Analysis (Dance-based) | Control Group from Meta-Analysis (vs. Dance-based) | Primary Outcome Measures Results from Meta-Analysis (Number of Studies) |
|---|---|---|---|---|---|---|---|---|---|---|---|
| Meiqi Zhang, et al. | 2023 | Exercise sustains motor function in Parkinson's disease: Evidence from 109 randomized controlled trials on over 4,600 patients | No | USA and China | 2 studies | 7 studies | Yes (Cochrane Collaboration's Tool for Assessing Risk of Bias) | PD | Tango, Tango or Waltz/Foxtrot, Partnered or non-partnered tango, Dance class (contemporary, ballet) + Physiotherapy, Partnered ballroom, NDT+FES+virtual reality dance, Turo Dance (Qi Dance) | Exercise, Physiotherapy, No intervention, Education, Wait-list, Dance class vouchers, NDT +FES | ↑ UPDRS III (δ), ↑ BBS (8), ↑TUG (δ), ↓ Purdue Pegboard Test (δ) |
| Di Wang, et al. | 2023 | Effectiveness of different exercises in improving postural balance among Parkinson's disease patients: a systematic review and network meta-analysis | No | China | 5 studies | 1 study (quasi-RCT) | Yes (Cochrane Risk of Bias tool) | PD | Sardinian Folk, Jazz, Waltz/Foxtrot or Tango, Tango, dance-physiotherapy, Irish dancing | Usual care, No intervention, Exercise group, conventional physiotherapy | ↑TUG (5), ↑ BBS (3), ⇆ Mini-BESTest (3) |
| Hayam Mahmoud Mahmoud, et al. | 2023 | Effect of dancing on freezing of gait in patients with Parkinson's disease: A systematic review and meta-analysis | Yes | Saudi Arabia and Egypt | 7 studies | 2 studies (pilot RCTs) | Yes (PEDro) | PD | Tango dance, Dancing physiotherapy, Irish dance, Tango with Waltz/Foxtrot, Educational lecture on dancing | Physiotherapy exercises, Regular physiotherapy exercise, No intervention | ⇆ nFOGQ (2), ⇆ FOGQ (7), ↑ UPDRS-III (9), ⇆ Mini-BESTest (4), ↑BBS (4), ⇆ TUG (8) |
| Shenglan He, et al. | 2023 | Whether mindfulness-guided therapy can be a new direction for the rehabilitation of patients with Parkinson's disease: a network meta-analysis of non-pharmacological alternative motor -/sensory-based interventions | No | China | 17 studies | No | Yes (Cochrane Risk of Bias tool) | PD | Tango, Samba and rhythmic dance | No intervention, exercise, Tai Chi, | ↑TUG (10), ↑ UPDRS (2), ⇆ UPDRS-I (5), ⇆ UPDRS-II (6), ↑ UPDRS-III (13), ⇆ PDQ-39 (9) |
| Patrícia Lorenzo-García, et al. | 2023 | Effects of physical exercise interventions on balance, postural stability and general mobility in Parkinson's disease: a network meta-analysis | No | Chile | 6 studies | No | Yes (Cochrane Collaboration Risk of Bias tool RoB2) | PD | Waltz/Foxtrot, Virtual reality dance exercise, Turo (Qi dance), Tango, Sardinian Folk dance, Irish dance group | No intervention, Usual care, Wait-list, Physiotherapy, Speed Treadmill Training, Mixed Treadmill, Latin Dance | ↑ BBS (4), ↑ TUG (3) |

*(Continued)*

Table 1. (Continued)

| Authors | Year | Title | Dance Specific | Country | RCT (Dance-based) | Non-RCT (Dance-based) | Risk of Bias Evaluation of Clinical Trials | Participants | Intervention Group from Meta-Analysis (Dance-based) | Control Group from Meta-Analysis (vs. Dance-based) | Primary Outcome Measures Results from Meta-Analysis (Number of Studies) |
|---|---|---|---|---|---|---|---|---|---|---|---|
| Sara Mohamed Hasan, et al. | 2022 | Efficacy of dance for Parkinson's disease: a pooled analysis of 372 patients | Yes | Egypt | 14 studies | No | Yes (Cochrane Risk of Bias tool) | PD | Tango, Waltz/Foxtrot, aerobic + tango + jazz + classic ballet, K-pop dance festival, dance therapy, Irish set dancing, Sardinian folk dance | Exercise, Physiotherapy or no intervention | ↑UPDRS III (11), ↑TUG (5), ↑BBS (3), ⇆ FOG (8), ⇆ 6mWt (5), ↑ MiniBEST (4), ⇆ BDI (2), ↑ AS (3), ↑ MOCA (3), ⇆ PDQ39 (9) |
| Claire Chrysanthi Karpodini, et al. | 2022 | Rhythmic cueing, dance, resistance training, and Parkinson's disease: A systematic review and meta-analysis | No | USA and Canada | 10 studies | 12 studies | Yes (Cochrane risk of bias tool for RCT e ROBINS-I tool for non-randomized controlled trials) | PD | Tango, Waltz/Foxtrot or Tango, Irish set dancing, Samba and Forro, Contemporary dance, Samba, Mark Morris Dance class for Parkinson, Mixed styles (jazz, tango, ballet, improvisation, and pantomime movements), Mixed styles (DFPD, Mark Morris ballet, jazz, tap, and movements for walking), Irish set dancing+ dance patterns such as Connemara Set, Kilfenora set, and Corofin plain set. + home dance, Mixed Ballroom (Waltz/foxtrot and tango) plus Latin American (cha cha, rock-and roll and rumba), Turo (Qi dance), Ballet, Dance (non-specified), DFPD, Seated and standing dance | Exercise, Physiotherapy, No intervention, Self-directed exercise, Usual care, Tai Chi, Not to be involved in dance classes, Physiotherapy+PD exercises or no intervention, Stretching or treadmill | ↑ UPDRS-III (13), ↑ TUG (11), ⇆ Gait velocity (8), ↑ Stride length (5), ⇆ PDQ39 (9), ⇆ MoCa (3) |

(Continued)

**Table 1.** (Continued)

| Authors | Year | Title | Dance Specific | Country | RCT (Dance-based) | Non-RCT (Dance-based) | Risk of Bias Evaluation of Clinical Trials | Participants | Intervention Group from Meta-Analysis (Dance-based) | Control Group from Meta-Analysis (vs. Dance-based) | Primary Outcome Measures Results from Meta-Analysis (Number of Studies) |
|---|---|---|---|---|---|---|---|---|---|---|---|
| Chun-Lan Yang, et al. | 2022 | Effects and parameters of community-based exercise on motor symptoms in Parkinson's disease: a meta-analysis | No | China | 3 studies | 6 studies | Yes (Cochrane tool) | PD | Tango, Tango or Waltz/Foxtrot, Community-based Tango, Sardinian folk dance, Dance therapy, Partnered ballroom dancing, Turo (Qi Dance), DFPD, Mixed types (jazz, tango, ballet, improvisation, and pantomime movements) | No intervention, Self-directed exercise, Support group, Dance class vouchers, Wait-list | ↑UPDRS-III (6) |
| Rustem Mustafaoglu, et al. | 2022 | Which type of mind–body exercise is most effective in improving functional performance and quality of life in patients with Parkinson's disease? A systematic review with network meta-analysis | No | Turkey and Hong Kong | 13 studies | No | Yes (Cochrane Risk of Bias assessment tool (Review Manager version 5.4.1) | PD | Tango, Sardinian folk dance, Waltz/ Foxtrot, Community-based Tango, Partnered or non-partnered dance, DFPD, Irish set dancing+video with steps, Mixed styles (aerobic, jazz, tango and ballet), NDT+FES+virtual reality dance, Irish set dancing and home dance, Ballroom (waltz/ foxtrot, and tango) and Latin American (cha cha, rock-and-roll and rumba) | Physiotherapy, Usual care, No intervention, Strength and flexibility exercises, NDT+FES | ↑ UPDRS III (8), ↑ TUG (9), ↑BBS (9)/BESTest (4), ↑10MWT (3), ↑ PDQ39 (6) |
| Patricia Lorenzo-Garcia, et al. | 2022 | Physical Exercise Interventions on Quality of Life in Parkinson's Disease: A Network Meta-analysis | No | Chile and Spain | 5 studies | No | Yes (Cochrane risk of bias tool RoB2) | PD | Tango, Irish set dancing, Waltz/ Foxtrot or Tango, Turo (qi dance), Latin dance +resistance band exercises, and coordination movements | Physiotherapy, No intervention, Usual care, Tai Chi, Speed treadmill, or Mixed treadmill | ↑ PDQ-39 (4) |

*(Continued)*

**Table 1.** (Continued)

| Authors | Year | Title | Dance Specific | Country | RCT (Dance-based) | Non-RCT (Dance-based) | Risk of Bias Evaluation of Clinical Trials | Participants | Intervention Group from Meta-Analysis (Dance-based) | Control Group from Meta-Analysis (vs. Dance-based) | Primary Outcome Measures Results from Meta-Analysis (Number of Studies) |
|---|---|---|---|---|---|---|---|---|---|---|---|
| Lina Goh, et al. | 2022 | The effect of rehabilitation interventions on freezing of gait in people with Parkinson's disease is unclear: a systematic review and meta-analyses | No | Australia | 3 studies | 4 studies | Yes (PEDro) | PD | Tango, Tango or Waltz/Foxtrot, Irish set dancing, Contemporary dance, Tango or Mixed styles therapeutic dance +home dance | Exercise, Physiotherapy, Usual care, Wait-list, No intervention, Self-directed exercise | ↑ NFOG-Q/ FOG-Q (3) |
| Yong Yang, et al. | 2022 | Efficacy and evaluation of therapeutic exercises on adults with Parkinson's disease: a systematic review and network meta-analysis | No | China | 10 studies | 1 study | Yes (TESTEX scale) | PD | Tango, partnered tango, Mixed styles, Irish dance | Exercise, another dance style, Dance for PD, Usual Care, Physiotherapy | ↑ Balance* (6), ↑ Depression* (δ) |
| Yuxin Wang, et al. | 2022 | Efficacy of non-pharmacological interventions for depression in individuals with Parkinson's disease: A systematic review and network meta-analysis | No | China | 3 studies | No | Yes (Cochrane) | PD | Tango, Sardinian folk dance, Turo (Qi dance) | Usual care, Wait-list, Self-directed exercise | ↓ BDI (2) / BDI-II (1) |
| Zikang Hao, et al. | 2022 | Effects of Ten Different Exercise Interventions on Motor Function in Parkinson's Disease Patients—A Network Meta-Analysis of Randomized Controlled Trials | No | China | 5 studies | 3 studies | Yes (Cochrane Handbook 5.1.0 tool) | PD | Tango, Tango or Waltz/Foxtrot, Sardinian Folk Dance, Irish set dancing, Rhythmic Auditory Stimulation | Exercise, Usual care, No intervention, Support group, Treadmill or Stretching, Self-directed exercise | ↑ UPDRS III (8), ↑TUG (4), ↑ BBS (4) |
| Li-li Wang, et al. | 2022 | Effects of dance therapy on non-motor symptoms in patients with Parkinson's disease: a systematic review and meta-analysis | Yes | China | 9 studies | 1 study | Yes (Cochrane Collaboration Risk of bias tool version 6.2) | PD | Tango, Sardinian Folk Dance, Turo (Qi Dance), Dance and physiotherapy, Mixed genres (jazz, tango, ballet), Open-air fitness dance, Virtual reality dance, Dance therapy (rhythmic movements, improvisation, free body expression) | Physiotherapy, Usual care, Self-directed exercise, Education, NDT +FES, Traditional talk therapy | ⇆ BDI (6), ⇆ FPS-16 (2), ⇆ AS (2), ↑ MOCA (6) |

(*Continued*)

Table 1. (Continued)

| Authors | Year | Title | Dance Specific | Country | RCT (Dance-based) | Non-RCT (Dance-based) | Risk of Bias Evaluation of Clinical Trials | Participants | Intervention Group from Meta-Analysis (Dance-based) | Control Group from Meta-Analysis (vs. Dance-based) | Primary Outcome Measures Results from Meta-Analysis (Number of Studies) |
|---|---|---|---|---|---|---|---|---|---|---|---|
| Celia Alvarez-Bueno, et al. | 2021 | Effect of exercise on motor symptoms in patients with Parkinson's Disease: a network meta-analysis | No | Spain, Paraguay, the United Kingdom, and Chile | 3 studies | 4 studies | Yes (Cochrane risk of bias tool RoB2) | PD | Tango, Waltz/Foxtrot or tango, Sardinian folk dance, Irish set dancing, Music therapy (rhythmic movements, improvisation, free body expression) | Exercise, Physiotherapy, Usual care | ↑UPDRS III (5) |
| Sophia Rasheeqa Ismail, et al. | 2021 | Evidence of disease severity, cognitive and physical outcomes of dance interventions for persons with Parkinson's Disease: a systematic review and meta-analysis | Yes | Malaysia | 20 studies | No | Yes (Cochrane Risk of Bias tool) | PD | Tango, Waltz/foxtrot or tango, Partnered or non-partnered Tango, DFPD or tango, Sardinian folk dance, Irish set dancing, Turo (Qi Dance), Ballroom and Latin American dance, Ballroom (Social Foxtrot, Waltz, Tango) and Latin American (Rumba, Cha Cha, Rock 'n' Roll), Dance Therapy, Tango or Mixed dance | Physiotherapy, Usual care, Education, Flexibility exercise, Stretching, Tai Chi, Tai Chi or no intervention, Wait-list, Support group, Not to be involved in dance classes | ↑UPDRS III (6), ⇆TUG (7), ↑MinBest (3), ⇆FOG-Q (3), ⇆6MWT (3), ↑FTSTT (1), ⇆SRT (1), ⇆Forward velocity (meters/second) (1), |
| Danique L. M. Radder, et al. | 2020 | Physiotherapy in Parkinson's Disease: A Meta-Analysis of Present Treatment Modalities | No | Netherlands | 11 studies | No | No | PD | Tango, Community-based tango, Partnered tango, Tango or Waltz/Foxtrot, Sardinian Folk Dance, Irish set dancing, Partnered or non-partnered dance, Mixzed types (jazz, tango, ballet), NDT+FES+virtual reality dance | Exercise, Physiotherapy, Usual care, No intervention, Self-directed exercise, NDT+FES | ↑UPDRS-III (8), ↑TUG (8), ↑BBS (7), ↑6MWT (4), ⇆Gait speed (6), ⇆Stride length (3), ↑Cadence (1), ⇆PDQ39 (3) |

(Continued)

**Table 1.** (Continued)

| Authors | Year | Title | Dance Specific | Country | RCT (Dance-based) | Non-RCT (Dance-based) | Risk of Bias Evaluation of Clinical Trials | Participants | Intervention Group from Meta-Analysis (Dance-based) | Control Group from Meta-Analysis (vs. Dance-based) | Primary Outcome Measures Results from Meta-Analysis (Number of Studies) |
|---|---|---|---|---|---|---|---|---|---|---|---|
| Maxwell Barnish & Barran | 2020 | A systematic review of active group-based dance, singing, music therapy, and theatrical interventions for quality of life, functional communication, speech, motor function, and cognitive status in people with Parkinson's disease | No | United Kingdom | 16 studies | 22 studies | Yes (Specialist Unit for Review Evidence checklist) | PD | Tango, Irish set dancing, Sardinian folk dance, Ballet, Modern dance, Turo (Qi Dance), DFPD, Tango or Waltz/ Foxtrot, Partnered and non-partnered tango, Dance therapy, Improvisational: seated at ballet Barre and ambulating, Dance program (improvisational and creative dance tasks.), PD-specific dance: alone/pair/group, Partnered dance ballroom and Latin, DfPD, and tango, Contact improvisation, Let's Dance, Mixed types (tap dancing, creative dance, and Irish dance) | Exercise, Usual care, Education, PD exercise, Wait-list, Not to be involved in dance classes, Support group, Treadmill or stretching Self-directed exercise | ⇆ UPDRS III (3), ↑ UPDRS III (2), ⇆ TUG (2), ↑ TUG (2), ↑ PDQ-39 (2) |
| Ruben D. Hidalgo-Agudo, et al. | 2020 | Additional Physical Interventions to Conventional Physical Therapy in Parkinson's Disease: A Systematic Review and Meta-Analysis of Randomized Clinical Trials | No | Spain | 2 studies | 1 study | Yes (PEDro) | PD | Partnered tango, Irish set dancing, Mixed styles (jazz, tango, ballet), | Strengthening exercise and core stability, Joint mobilization, balance and walking training by video or book or no intervention, Self-directed exercise | ↑ UPDRS III (2), ↑ TUG (2), ↑ BBS (2) |
| Anna M. Carapellottil, et al. | 2020 | The efficacy of dance for improving motor, non-motor symptoms, and quality of life in Parkinson's disease: A systematic review and meta-analysis | Yes | United Kingdom | 16 studies | No | Yes (Cochrane Risk of Bias tool) | PD | Tango, Tango or Waltz/Foxtrot, Partnered or nonpatterned Tango, Tango or Mixed styles, Irish set dancing, Sardinian folk dance, Turo PD/ Qi dance, Ballroom/ Latin American, Dance Therapy | Exercise, No intervention, Usual Care, Support group, Tai Chi or no intervention | ↑ UPRS III (4), ↑ TUG (3), ⇆ FOGQ (2), Forward gait velocity (3), ↑ 6MWT (4), ↑ BDI-II (1), ⇆ Backward gait velocity (2), ⇆ Forward stride length (2), ⇆ PDQ-39 (3) |

*(Continued)*

**Table 1.** (Continued)

| Authors | Year | Title | Dance Specific | Country | RCT (Dance-based) | Non-RCT (Dance-based) | Risk of Bias Evaluation of Clinical Trials | Participants | Intervention Group from Meta-Analysis (Dance-based) | Control Group from Meta-Analysis (vs. Dance-based) | Primary Outcome Measures Results from Meta-Analysis (Number of Studies) |
|---|---|---|---|---|---|---|---|---|---|---|---|
| Kui Chen, et al. | 2020 | Effect of Exercise on Quality of Life in Parkinson's Disease: A Systematic Review and Meta-Analysis | No | China | 1 study | 2 studies | Yes (Cochrane Risk of Bias tool) | PD | Tango, Tango or Waltz/Foxtrot, DFPD | Education, Tai Chi, Support group | ↑ PDQ-39 (3) |
| Heloisa de Almeida, et al. | 2020 | Effect of Dance on Postural Control in People with Parkinson's Disease: A Meta-Analysis Review | Yes | Brazil | 4 studies | 8 studies | Yes (Cochrane Collaboration tool) | PD | Tango, Tango or Waltz/Foxtrot, Community-based tango, Sardinian Folk Dance, Turo (Qi Dance), Irish set dancing + home program, Mixed styles (jazz, tango, ballet), NDT+FES +virtual reality dance, Partnered dance, Dance Therapy | Exercise, No intervention, Physiotherapy, Self-directed exercise, NDS +FES, Wait-list, Dance class vouchers, Usual care, Support group | ↑ BBS (11) / MiniBEST (2) |
| Nadeesha Kalyani H. Haputhanthirigea, et al. | 2019 | Effects of Dance on Gait, Cognition, and Dual-Tasking in Parkinson's Disease: A Systematic Review and Meta-Analysis | Yes | Australia | 9 studies | 18 studies | Yes (Cochrane database) | PD | Tango, Irish set dancing, Contact improvisation dance, DFPD, Thai classical dance, Mixed styles (jazz, tango, ballet), Dance (Not specified) | Exercise, No intervention, Home-based exercises, Education | ↑ Gait velocity (m/s) (6), ↑ Dual-task TUG (3), ↑ MoCa (2) |
| Lijun Tang, et al. | 2019 | The effects of exercise interventions on Parkinson's disease: A Bayesian network meta-analysis | No | China | 3 studies | 2 studies | Yes (Cochrane tool) | PD | Tango, Tango or Waltz/Foxtrot, partnered tango, Mixed Types (jazz, tango, ballet), Dance Therapy | Exercise or no intervention, No intervention, Self-directed exercise, Support group | ↑ UPDRS III (4), ↑ TUG (3), ⇆ Gait velocity (1), ⇆ PDQ-39 (3), ↑ BBS (4) |
| Qi Zhang, et al. | 2019 | Effects of dance therapy on cognitive and mood symptoms in people with Parkinson's disease: A systematic review and meta-analysis | Yes | China | 7 studies | No | Yes (Cochrane database) | PD | Tango, Mixed types (jazz, tango, ballet), Turo (Qi dance), NDT+FES+Virtual reality dance, Dance Therapy | Exercise, Education, Self-directed exercise, NDT+FES | ⇆ MoCA (2), ↑ FAB (2), ↓ SDS, ⇆ BDI (4), ⇆ AS (2) |

*(Continued)*

Table 1. (Continued)

| Authors | Year | Title | Dance Specific | Country | RCT (Dance-based) | Non-RCT (Dance-based) | Risk of Bias Evaluation of Clinical Trials | Participants | Intervention Group from Meta-Analysis (Dance-based) | Control Group from Meta-Analysis (vs. Dance-based) | Primary Outcome Measures Results from Meta-Analysis (Number of Studies) |
|---|---|---|---|---|---|---|---|---|---|---|---|
| Camila Monteiro Mazzarin, et al. | 2017 | Effects of Dance and of Tai Chi on Functional Mobility, Balance, and Agility in Parkinson's Disease—A Systematic Review and Meta-analysis | No | Brazil | 5 studies | No | Yes (Jadad scale) | PD | Tango, Ronnie Gardiner Rhythm, Irish set dancing | Exercise, No intervention, Physiotherapy | ⇆ UDPRS III (3), ⇆ BBS (3) |
| Marcela Delabary, et al. | 2017 | Effects of dance practice on functional mobility, motor symptoms and quality of life in people with Parkinson's disease: a systematic review with meta-analysis | Yes | Brazil | 3 studies | 2 studies | Yes (Cochrane database) | PD | Partnered tango classes, Irish set dancing, Tango, Community-based Tango, Tango or Waltz/Foxtrot | Exercise, Self-directed exercise, No intervention | ↑ UPDRS III (3), ↑TUG (2), ⇆ FOGQ (3), ⇆ PDQ39 (2). Dance x no intervention: ↑ UPDRS III (2), ⇆ 6MWT (2), ⇆ FOGQ (2), ⇆ Forward velocity (2), ⇆ backward velocity (2) |
| Kwok Yan Yan, et al. | 2016 | Effects of mind-body exercises on the physiological and psychosocial well-being of individuals with Parkinson's disease: A systematic review and meta-analysis | No | Hong Kong | 2 studies | 1 study | Yes (Effective Public Health Practice Project quality assessment tool) | PD | Tango, Waltz/Foxtrot, Modern dance | No intervention, Tai Chi | ↑ UPDRS III (2), ↑ BBS (2), ↑ TUG (2), ↓ 6MWT (1) |
| Désirée Lötzke, et al. | 2015 | Argentine tango in Parkinson's disease—a systematic review and meta-analysis | Yes | Germany | 9 studies | 4 studies | Yes (Authors Criteria) | PD | Tango, Partnered tango, Community-based tango, Tango or Waltz/Foxtrot, Tango + tDCS or Tango + Sham | Exercise, No intervention, Wait-list, Tai Chi | ↑ UPDRS III (3), ↑ Mini-BESTesT (3) ↑ BBS (3), ↑TUG (6), ↓ 6MWT (4), ↑ FOGQ (4) |
| Joanne Shanahan, et al. | 2015 | Dance for People with Parkinson's Disease: What Is the Evidence Telling Us? | Yes | Ireland | 8 studies | 1 study | Yes (PEDro) | PD | Tango, Irish set dancing, Community-based tango, Partnered or non-partnered tango, Tango or Waltz/Foxtrot | Exercise, No intervention, Physiotherapy, Education, Tai Chi | ↑ UPDRS III (5), ↑ BBS (5), ↑TUG (5) |

(Continued)

**Table 1.** (Continued )

| Authors | Year | Title | Dance Specific | Country | RCT (Dance-based) | Non-RCT (Dance-based) | Risk of Bias Evaluation of Clinical Trials | Participants | Intervention Group from Meta-Analysis (Dance-based) | Control Group from Meta-Analysis (vs. Dance-based) | Primary Outcome Measures Results from Meta-Analysis (Number of Studies) |
|---|---|---|---|---|---|---|---|---|---|---|---|
| Priscila A. da Rocha, et al. | 2015 | Complementary physical therapies for movement disorders in Parkinson's disease: a systematic review | No | Brazil and Australia | 6 studies | 3 studies | Yes (Cochrane database) | PD | Tango, Tango or Waltz/Foxtrot, Irish set dancing Partnered or non-partnered dance, Contact improvisation | Exercise, No intervention, Tai Chi or No intervention | ↑ 6MWT (2), ↑ BBS (3), ↑ UPDRS III (4) |
| Kathryn Sharp & Hewitt | 2014 | Dance as an intervention for people with Parkinson's disease: A systematic review and meta-analysis | Yes | United Kingdom | 5 studies | No | Yes (Cochrane tool) | PD | Tango, Tango or Waltz/Foxtrot and Irish set dancing | Exercise, No intervention, Physiotherapy, Tai Chi or no intervention | ↑ UPDRS (2), ↑ BBS (2), ⇆ FOGQ (2), ⇆ Velocity (m/s) (2), ⇆ 6MWT (2), ↑ PDQ-39 (2) |
| M. J. de Dreu, et al. | 2012 | Rehabilitation, exercise therapy and music in patients with Parkinson's disease: a meta-analysis of the effects of music-based movement therapy on walking ability, balance, and quality of life | Yes | Netherlands | 7 studies | No | Yes (PEDro) | PD | Tango, Waltz/Foxtrot, Music Therapy, RAS, walking program with music | Physiotherapy, Usual care, Education, Flexibility exercise, Stretching, Tai Chi, no intervention, Wait-list, Support group, not to be involved in dance classes | ↑ BBS (2), ↑TUG (2), ↑ Stride Length (3), ⇆ Walking Velocity (4), ⇆ UPDRS-III (4) ⇆ FOGQ (2) |
| Claire L. Tomlinson, et al. (A) | 2012 | Physiotherapy versus placebo or no intervention in Parkinson's (Cochrane Review) | No | United Kingdom | 2 studies | No | Yes (Authors Criteria) | PD | Tango, Waltz/Foxtrot (partnered) | No intervention | ↑ UPDRS III (2), ⇆ TUG (1), ⇆ BBS (1) |
| Claire L. Tomlinson, et al. (B) | 2012 | Physiotherapy intervention in Parkinson's disease: systematic review and meta-analysis | No | United Kingdom | 2 studies | No | Yes (Authors Criteria) | PD | Tango, Waltz/Foxtrot (partnered) | No intervention | ↑ UPDRS III (2), ⇆ TUG (1), ⇆ BBS (1) |

*Note.* The table represents the characteristics of each systematic review. Columns named RCT, non-RCT, Intervention group, Control group, and Primary outcome measures refer only to dance-based intervention studies included in each review.

*Symbols:* '↑' dance is superior to control, '⇆' dance is equal to control, '↓' dance is inferior to control, '⁕' outcome measurements not clear, 'δ' not clear or not provided, '$' conflicting data.

*Abbreviations.* AS (Apathy Scale). BBS (Berg Balance Scale). BDI (Beck Depression Inventory). CNKI (China National Knowledge Infrastructure. Cochrane CENTRAL (Cochrane Central Register of Controlled Trials). DFPD (Dance for Parkinson's Disease). FAB (Frontal Assessment Battery). FES (Functional Electrical Stimulation). FPS-16 (Parkinson's Fatigue Scale). GRADE (Grading of Recommendations, Assessment, Development, and Evaluations). MOCA (Montreal Cognitive Assessment Scale). NDT (Neurodevelopment). PRISMA (Preferred Reporting Items for Systematic Reviews and Meta-Analyses). RAS (Rhythmic Auditory Stimulation). RCT (randomized controlled trial). SDS (Self-rating Depression Scale). SRT (Sit-and-Reach Test). tDCS (transcranial direct current stimulation). TUG (Timed Up and Go Test) to investigate functional mobility. TURO PD (combination of Qigong and meditation to be danced with music). UPDRS III (Unified Parkinson Disease Rating Scale, Motor Section). WOS (Web of Science). WHO: World Health Organization International Clinical Trials Registry Platform. 6MWT (six-minute walk test).

**Table 2. Characteristics of included reviews without MA.**

| Authors | Year | Title | Dance Specific | Country | RCT (Dance-based) | Non-RCT (Dance-based) | Risk of Bias Evaluation of Clinical Trials | Participants | Intervention Group (Dance-based) | Control Group (vs. Dance-based) | Primary Outcome Measures (Number of Studies) |
|---|---|---|---|---|---|---|---|---|---|---|---|
| Valton Costa, et al. | 2023 | Physical exercise for treating non-motor symptoms assessed by general Parkinson's disease scales: systematic review and meta-analysis | No | Brazil | 4 studies | No | Yes (PEDro) | PD | Binary rhythm dance, Quaternary rhythm dance, Dance-therapy (multi-type), Turo (Qi dance), Tango | Support group, No intervention, Usual routine | ⇆ MDS-UPDRS I (2), ⇆ UPDRS-I (2), |
| Raluca-Dana Mot & Almăjan-Guță | 2022 | Dance therapy for Parkinson's disease: a systematic review | Yes | Romania | 6 studies | 14 studies | No | PD | Tango, Tango +education, Dance group, D4PD, Tango Group, Dance Therapy, Dance class, Tap, Irish and creative dance steps, Irish dance, Ballet, Jazz, Broadway-style dance, NDT+FES +dance (KPop, Wii, Nintendo), Improvisation dance, Dance Revolution (Wii) | No intervention, NDT+FES, Physiotherapy, Self-directed exercise at home daily | BBS (6), TUG (5), 6MWT (2), UPDRS-III (4), BDI (3) |
| Riddhi Dipak Patel & Mans | 2022 | Review: Effect of Supplemental Activities on Motor and Nonmotor Outcomes in the Parkinson's Population | No | United States | 8 studies | No | Yes (PEDro) | PD | Choreography from several dance styles, Folk dance, continuously dancing with partners | No intervention, cardiovascular program, another exercise, or usual care. | TUG (2), BBS (3), MiniBEST (2), 6MWT (6), Gait parameters (2), FOGQ (2), UPDRS (3). |
| Cheng-Cheng Wu, et al. | 2022 | Dance movement therapy for neurodegenerative diseases: A systematic review | Yes | China | 18 studies | No | Yes (PEDro and AMSTAR) | PD, Alzheimer's disease, Cognitive impairment | Tango Group/ Partnered, Tango, Waltz/Foxtrot, Irish set dancing, Sardinian folk dance | Usual care, Physiotherapy, Multimodal exercise, another dance style | UPDRS Total (4), UPDRS-I (3), UPDRS-II (1), UPDRS-III (11), TUG (8), Dual-task TUG (4), BBS (7), FOG_Q (4), New FOGQ (1), FOG (2), 6MWT (6), Mini-BESTest (5), PDQ-39 (5), FTSST (1), MIMUs (1), MOCA (6), BDI (5), BDI-II (1), BDI-21 (1), SAS (1), AS (2), FSS (2), Falls (1), BMLSS (1), VAFS (1), FAB (1), MRT (1), SDS (1), MMSE (1), MBI (1), ACS (1), 9HPT (1), ADL (1), H&Y (2), Gait parameters with 3D Motion Analysis (1), Gait parameters with GAITRite (3), Twelve, 180° on-the-spot turns (1) |

*(Continued)*

**Table 2.** (Continued)

| Authors | Year | Title | Dance Specific | Country | RCT (Dance-based) | Non-RCT (Dance-based) | Risk of Bias Evaluation of Clinical Trials | Participants | Intervention Group (Dance-based) | Control Group (vs. Dance-based) | Primary Outcome Measures (Number of Studies) |
|---|---|---|---|---|---|---|---|---|---|---|---|
| Sara Emmanouilidis, et al. | 2021 | Dance Is an Accessible Physical Activity for People with Parkinson's Disease | Yes | Australia | 17 studies | Yes (21 studies) | Yes (PEDro) | PD | Tango, Sardinian folk dancing, Irish set dancing, Waltz/foxtrot, Dance (virtual reality dance, Dance google Glass modules, Ballet, Brazilian samba, Zumba, Qigong dance, Improvisation dance, Mixed dance genres | Usual care, Usual physical activity, and Other therapeutic intervention | UPDRS Total (7), UPDRS-I (3), UPDRS-II (4), UPDRS-III (14), FOG (2), FOGQ (8), 6MWT (14), MiniBESTest (8), TUG (19), dual-task TUG (5), BBS (15), SS180 (1), Phone-FITT (1), EQ-5D (2), ABC (5), 5TSTST (1), SDS (1), FAB (3), MRT (1), AS (3), BDI (9), MBI (1), PDQL (2), FSST (1), PDQ-39 (10), SF-12 (2), MoCA (4), FSS (2), VAS-F (1), BMLSS (1), ICPH (1), GSE-6 (1), QOLS (1), PFS-16 (1), SAS (1), MDMQ (1), HADS (1), FGA (1), KFSS (1), CGI-C (1), IPAQ (1), DGI (1), SF-36 (1), Falls Questionnaire (1), Heart rate and rated perceive exertion (1), CEDS Depression Scale (1), Westheimer Questionnaire (1), Tinetti POMA Test (1), Self-Efficacy Scale (1), Fear of falling (1), Gait parameters with GAITRite (9), Gait parameters with 3D Motion Analysis (2), Cadence (1)—Using stopwatch over a 20' path, 30-second chair stand (1), Tandem Stance Test (2), One Leg Stance Test (2) |
| A. Berti, et al. | 2020 | Argentine tango in the care of Parkinson's disease: A systematic review and analysis of the intervention | Yes | Italy | 10 studies | 10 studies | Yes (PEDro and MINORS) | PD | Tango | Not informed | NA |

(*Continued*)

**Table 2.** (Continued)

| Authors | Year | Title | Dance Specific | Country | RCT (Dance-based) | Non-RCT (Dance-based) | Risk of Bias Evaluation of Clinical Trials | Participants | Intervention Group (Dance-based) | Control Group (vs. Dance-based) | Primary Outcome Measures (Number of Studies) |
|---|---|---|---|---|---|---|---|---|---|---|---|
| Adijatu Raheem & Casaca-Carreira | 2018 | Effects of ballroom dancing in patients with Parkinson's disease: A systematic review | Yes | Portugal | 7 studies | No | No | PD | Tango and Waltz/ foxtrot | Physical exercise, No intervention, Tai Chi | UPDRS-III (5), UPDRS-I (2), Mini BESTest (3), BBS (2), FOG (5), 6MWT (3), MOCA (1), BDI (1), AS (1), KFSS (1), FRT (1), TUG (4), Dual-task TUG (2), One leg Stance Test (1), Gait parameters with GAITRite (3), Gait parameters with 3D Motion Analysis (1), |
| Lorenna Pryscia C. Aguiar, et al. | 2016 | Therapeutic Dancing for Parkinson's Disease | Yes | Australia | 9 studies | 10 studies | Yes (PEDro and Downs and Black) | PD | Tango, Waltz/ Foxtrot, Contemporary dance, Irish set dancing, Modern dance, Ballet | Control, Exercise, another dance style | UPDRS III (13), UPDRS I (1), UPDRS (1), PDQ-39 (6), BBS (9), FOGQ (6), TUG (10), Mini-BESTest (1), 6MWT (6), GAITRite (5), PASE (1), FAB (2), FSST (1), ABC (1), FTSTS (1), One Leg Stance Test (1), Gait parameters with 3D Motion Analysis (1), Euroqol5D (1), Modified Westheimer QOLS (1), Semi-tandem test (1) |
| Melanie E. Cusso, et al. | 2016 | The Impact of Physical Activity on Non-Motor Symptoms in Parkinson's Disease: A Systematic Review | No | Australia | 2 studies | No | Yes (Cochrane Collaboration's tool) | PD | Tango | Usual care, educational material | UPDRS I (1), MOCA (1), BDI (1), KFSS (1), AS (1) |
| Luís A. A. Santos, et al. | 2016 | Effects of dual-task interventions on gait performance of patients with Parkinson's Disease: A systematic review | No | Brazil and Portugal | 1 study | 2 studies | No | PD | Tango | Daily home-based exercises, No exercise | TUG (1), Dual-Task TUG (2), FOGQ (1), 6MWT (2), Gait Speed with GAITRite (1) |
| Rastilav Šumec, et al. | 2015 | Psychological Benefits of Nonpharmacological Methods Aimed for Improving Balance in Parkinson's Disease: A Systematic Review | No | Czech Republic and USA | 2 studies | 10 studies | | PD | Tango, Waltz/ Foxtrot, Partnered tango, Contact improvisation, Modern dance, Ballet, Dance not specified, D4PD | No intervention | UPDRS Total (1), UPDRS III (4), BBS (6), 6 MWT (1), FAB (3), TUG (5), QoL (1), ABC (1), Modified FaES (1), FRT (1), UST (1), Mini-BESTest (1), FSS (1), SeTa (1), Gait parameters with GAITRite (4), Gait parameters with 3D Motion Analysis (1) |

(*Continued*)

**Table 2.** (Continued)

| Authors | Year | Title | Dance Specific | Country | RCT (Dance-based) | Non-RCT (Dance-based) | Risk of Bias Evaluation of Clinical Trials | Participants | Intervention Group (Dance-based) | Control Group (vs. Dance-based) | Primary Outcome Measures (Number of Studies) |
|---|---|---|---|---|---|---|---|---|---|---|---|
| Rosalind Mandelbaum & Lo | 2014 | Examining Dance as an Intervention in Parkinson's Disease: A Systematic Review | Yes | United States | 3 studies | 7 studies | No | PD | Tango, Tango Partnered, Modern dance, Argentine Tango, Waltz/Foxtrot, D4PD, Contact Improvisation Partnered, Dance Therapy | No comparator, another dance style | UPDRS (4), UPDRS-III (1), TUG (6), BBS (5), 6MWT (3), FOGQ (1), FAB (1), PGCM (1), Mini BESTest (1), Gait parameters with 3D Motion Analysis (2), Gait parameters with GAITRite (4), Functional ambulation profile (1) |
| Danielle K. Murray, et al. | 2014 | The effects of exercise on cognition in Parkinson's disease: a systematic review | No | Canada | 1 study | No | No | PD | Tango, moderate-to-high intensity anabolic and aerobic exercise, low-intensity passive aerobic exercise, moderate-intensity multimodal exercise training, Wii Fit program, a single bout of high-intensity endurance aerobic exercise—heart rate-targeted cycling, low-intensity aerobic exercise program with Nordic walking poles—Pole Striding | Education lessons | UPDRS-III (1), Brooks Spatial Task (1) |
| E. Valverde Guijarro & García | 2012 | Efecto de la danza en los enfermos de Parkinson | Yes | Spain | 6 studies | 7 studies | Yes (PEDro) | PD | Tango, Tango, Waltz/Foxtrot, Modern Dance, Classic ballet, Jazz, Contemporary Dance, Theatre dance, Choreography, Contact improvisation, Couple dance, Dance therapy | No intervention, No control, Flexibility exercise, Exercise +dance movement therapy, Respiratory exercises, Stretching, Resistance, Dexterity, and another dance style | UPDRS (10), PDQ-39 (3), BBS (6), ABC (3), FAB (1), Mini BESTest (1), FOGQ (4), FaES (1), BDI (1), TUG (7), 6MWT (6), QoLs (1), FRT (2), FTSTS (1), 9HPT (1), PGCM (1), Gait parameters with 3D Motion Analysis (2), Gait parameters with GAITRite (2), One Leg Stance Time Test (2), Tandem Stance Time (1), SeTa (1), Westheimer (1) |

(*Continued*)

**Table 2.** (Continued)

| Authors | Year | Title | Dance Specific | Country | RCT (Dance-based) | Non-RCT (Dance-based) | Risk of Bias Evaluation of Clinical Trials | Participants | Intervention Group (Dance-based) | Control Group (vs. Dance-based) | Primary Outcome Measures (Number of Studies) |
|---|---|---|---|---|---|---|---|---|---|---|---|
| Marie-Sophie Kiepe, et al. | 2012 | Effects of dance therapy and ballroom dances on physical and mental illnesses: A systematic review | Yes | Germany | 2 studies | No | No | PD, Cancer, Dementia, Heart Failure, Depression, Diabetes type 2, and Fibromyalgia | Argentine Tango and Waltz/ Foxtrot | Strength/ flexibility exercises, Tai Chi, and No intervention | UPDRS-III (1), BBS (2), FOGQ (1), 6MWT (1), PDQ-39 (1) |

*Note.* Tables 1, 2 represents the characteristics of each systematic review. Columns named RCT, non-RCT, Intervention group, Control group, and Primary outcome measures refer only to dance-based intervention studies included on each review.

*Symbols*: '↑' dance is superior to control, '⇆' dance is equal to control, '↓' dance is inferior to control, '*' outcome measurements not clear, 'δ' not clear or not provided, '$' conflicting data

*Abbreviations.* ABC (Activities-Specific Balance Confidence Scale). ACS (Activity Card Sort). ADL (Activities of Daily Living). AMED (Allied and Complementary Medicine Database). AS (Apathy Scale). BBS (Berg Balance Scale). BDI (Beck Depression Inventory). BMLSS (Body Mass Lower and Upper Limb). CENTRAL (Cochrane Central Register of Controlled Trials). CGI-C (Clinical Global Impression-Change). CINAHL (Cumulative Index to Nursing and Allied Health Literature). CNKI (China National Knowledge Infrastructure). Cochrane CENTRAL (Cochrane Central Register of Controlled Trials). D4PD (Dance for Parkinson's Disease). DFPD (Dance for Parkinson's Disease). DGI (Dynamic Gait Index). DTF (Difficulty Task Fatigue). EBM Reviews (Evidence-Based Medicine Reviews). EMBASE (Excerpta Medica Database). EQ-5D (EuroQol 5-Dimension Scale). FAB (Frontal Assessment Battery). FaES (Falls Efficacy Scale). FES (Functional Electrical Stimulation). FGA (Functional Gait Assessment). FOGQ (Freezing of Gait Questionnaire). FPS-16 (Parkinson's Fatigue Scale). FRT (Functional Reach Test). FSS (Flow State Scale). FSST (Four-Square Step Test). FTSST (Five Times Sit to Stand Test). GRADE (Grading of Recommendations, Assessment, Development, and Evaluations). GSE-6 (Short Scale Of General Self-Efficacy). HADS (Hospital Anxiety and Depression Scale). H&Y (Hoehn and Yahr Scale). ICPH (Inner Congruence with Practices). IPAQ (Impact on Participation and Autonomy Questionnaire). KFSS (Rupp's Fatigue Severity Scale). MBI (Mild Behavioural Impairment). MDMQ (Multidimensional Mood State Questionnaire). MIMUs (Miniaturized Inertial Measurement Units). Mini-BESTest (Mini-Balance Evaluation Systems Test). MMSE (Mini-Mental State Examination). MOCA (Montreal Cognitive Assessment Scale). MRT (Metabolic Rate Test). NDT (Neurodevelopment). NDT+FES (Neurodevelopment+Functional Electrical Stimulation). NA (Not Applicable). PASE (Physical Activity Scale in the Elderly). PDQ-39 (The 39-item Parkinson's Disease Questionnaire). PDQL (Parkinson's Disease Quality of Life). PEDro (Physiotherapy Evidence Database). PFS-16 (Parkinson's Disease Fatigue Scale). PGCM (Philadelphia Geriatric Center Morale). PRISMA (Preferred Reporting Items for Systematic Reviews and Meta-Analyses).QOLS (Quality of Life Scale). RAS (Rhythmic Auditory Stimulation). RCT (randomized controlled trial). SAS (Starkstein Apathy Scale). SDS (Self-Rating Depression Scale). SeTa (Sitting and Tandem). SF-12 (12-item Short-Form Health Survey). SF-36 (36-Item Short Form Health Survey). SRT (Sit-and-Reach Test). SS180 (Standing-start 180˚). tDCS (transcranial direct current stimulation). TIDieR (Template for Intervention Description and Replication). TMT A&B (Trail Making Test A and B). TUG (Timed Up and Go Test). TURO PD (combination of Qigong and meditation to be danced with music). UPDRS (Unified Parkinson Disease Rating Scale). UPDRS-I (Unified Parkinson Disease Rating Scale, Section Cognitive). UPDRS-II (Unified Parkinson Disease Rating Scale, Section Daily Activities). UPDRS III (Unified Parkinson Disease Rating Scale, Motor Section). UST (Upper-Body Strength Training. VAFS (VaultTec Accelerated Focus System). WHO (World Health Organization International Clinical Trials Registry Platform). WOS (Web of Science). 6MWT (six-minute walk test). 9HPT (9-Hole Peg Test).

*Rios Romenets et al. (2015)* [85] differed from the other RCTs because its control group associates pharmacological usual care with a self-administered learning intervention (i.e. participants received a pamphlet about exercises for people with PD from *Parkinson Society of Canada* and were instructed to practice the exercises at home [85]). Another study included three groups and compared mixed repertoire dance in one program to pharmacological usual care or an active multimodal exercise program [88]. Another study compared dance (Irish dance) to an active multimodal exercise program [89]. Participants included in experimental groups with dance maintained their pharmacological usual care (i.e. antiparkinsonian drugs) during the programs. All RCTs included in meta-analyses used the same test to assess outcomes of interest, thereby substantially reducing the heterogeneity of our results. Thus, we included seven RCTs for the meta-analyses.

**Table 3. AMSTAR 2.0.**

| Author / Year | 1 | 2 | 3 | 4 | 5 | 6 | 7 | 8 | 9 | 10 | 11 | 12 | 13 | 14 | 15 | 16 | Overall Score |
|---|---|---|---|---|---|---|---|---|---|---|---|---|---|---|---|---|---|
| Wei-Hsin Cheng, et al. 2024 | Y | N | N | PY | Y | Y | PY | PY | Y for RCT / only RCT | N | Y for RCT / N MA for NRSI | Y | Y | Y | Y | Y | Low |
| Ernst M., et al. 2024 | Y | Y | Y | Y | Y | Y | Y | Y | Y for RCT / only RCT | Y | Y for RCT / N MA for NRSI | Y | Y | Y | Y | Y | High |
| Donida G., et al. 2023 | Y | Y | Y | PY | Y | Y | N | PY | Y for RCT / only RCT | N | Y for RCT / N MA for NRSI | Y | Y | Y | Y | Y | Low |
| Caroline Simpkins & Yang 2023 | Y | N | Y | PY | Y | Y | Y | PY | Y for RCT / only RCT | N | Y for RCT/ N MA for NRSI | Y | Y | Y | Y | Y | Low |
| Meiqi Zhang, et al. 2023 | Y | Y | Y | PY | N | Y | Y | N | Y for RCT / only RCT | N | Y for RCT / N MA for NRSI | Y | Y | N | Y | Y | Moderate |
| Di Wang, et al. 2023 | Y | Y | Y | PY | Y | N | PY | PY | Y for RCT / N for NRCT | Y | Y for RCT / N MA for NRSI | Y | Y | Y | Y | Y | Low |
| Hayam Mahmoud Mahmoud, et al. 2023 | N | Y | Y | PY | N | N | PY | PY | Y for RCT / N for NRCT | N | Y for RCT / N MA for NRSI | Y | N | Y | N | Y | Critically Low |
| Shenglan He, et al. 2023 | Y | Y | Y | PY | Y | Y | N | PY | Y for RCT / only RCT | N | Y for RCT / N MA for NRSI | Y | N | Y | N | Y | Critically Low |
| Patrícia Lorenzo-García, et al. 2023 | Y | Y | Y | PY | Y | Y | PY | PY | Y for RCT / only RCT | N | Y for RCT / N MA for NRSI | Y | Y | Y | Y | Y | High |
| Valton Costa, et al. 2023 | Y | N | Y | PY | Y | Y | PY | PY | Y for RCT / only RCT | N | N MA | N MA | Y | Y | N MA | Y | Low |
| Claire Chrysanthi Karpodini, et al. 2022 | Y | Y | Y | N | Y | Y | Y | Y | Y for RCT / Y for NRSI | N | Y for RCT / N MA for NRSI | Y | Y | N | Y | Y | Critically Low |
| Cheng-Cheng Wu, et al. 2022 | Y | Y | Y | PY | Y | Y | N | PY | Y for RCT / only RCT | N | N MA | N MA | N | N | N MA | Y | Critically Low |

(*Continued*)

**Table 3.** (Continued)

| Author / Year | 1 | 2 | 3 | 4 | 5 | 6 | 7 | 8 | 9 | 10 | 11 | 12 | 13 | 14 | 15 | 16 | Overall Score |
|---|---|---|---|---|---|---|---|---|---|---|---|---|---|---|---|---|---|
| Chun-Lan Yang, et al. 2022 | Y | Y | Y | PY | N | Y | Y | PY | Y for RCT / Y for NRSI | N | N for RCT / N for NRSI | Y | N | Y | Y | Y | Critically Low |
| Lina Goh, et al. 2022 | Y | Y | Y | PY | Y | Y | Y | Y | Y for RCT / only RCT | N | Y for RCT / N MA for NRSI | Y | Y | Y | Y | Y | High |
| Patricia Lorenzo-García, et al. 2022 | Y | Y | Y | N | N | N | Y | PY | Y for RCT / only RCT | N | Y for RCT / N MA for NRSI | Y | Y | N | N | Y | Critically Low |
| Rustem Mustafaoglu, et al. 2022 | Y | Y | Y | PY | Y | Y | Y | Y | Y for RCT / only RCT | N | Y for RCT / N MA for NRSI | Y | Y | Y | Y | Y | High |
| Raluca-Dana Mot & Almăjan-Guţă 2022 | N | N | Y | PY | N | N | N | N | N for RCT / only RCT | N | N MA | N MA | N | Y | N MA | N | Critically Low |
| Riddhi Dipak Patel & Mans 2022 | N | N | Y | PY | N | N | PY | N | Y for RCT / only RCT | N | N MA | N MA | Y | N | N MA | Y | Low |
| Yong Yang, et al. 2022 | Y | Y | Y | PY | Y | Y | Y | Y | Y for RCT / only RCT | N | Y for RCT / N MA for NRSI | Y | Y | Y | Y | Y | High |
| Yuxin Wang, et al. 2022 | Y | N | Y | PY | N | Y | Y | PY | Y for RCT / only RCT | N | Y for RCT/ N MA for NRSI | Y | Y | Y | Y | Y | Low |
| Zikang Hao, et al. 2022 | Y | N | Y | N | Y | N | Y | PY | Y for RCT / only RCT | N | Y for RCT / N MA for NRSI | N | N | N | Y | Y | Critically Low |
| Celia Alvarez-Bueno, et al. 2021 | Y | Y | Y | PY | Y | Y | N | PY | Y for RCT / Y for NRSI | N | N for RCT / N MA for NRSI | Y | Y | Y | Y | Y | Critically Low |
| Sara Emmanouilidis, et al. 2021 | Y | N | Y | PY | Y | Y | Y | Y | Y for RCT / N for NRSI | N | N MA | N MA | Y | Y | N MA | Y | Critically Low |
| Sara Mohamed Hasan, et al. 2021 | Y | N | Y | PY | Y | Y | Y | Y | Y for RCT / N for NRSI | N | Y for RCT / N MA for NRSI | N | N | N | N | Y | Critically Low |

(*Continued*)

**Table 3.** (Continued)

| Author / Year | 1 | 2 | 3 | 4 | 5 | 6 | 7 | 8 | 9 | 10 | 11 | 12 | 13 | 14 | 15 | 16 | Overall Score |
|---|---|---|---|---|---|---|---|---|---|---|---|---|---|---|---|---|---|
| Sophia Rasheeqa Ismail, et al. 2021 | Y | Y | Y | PY | Y | Y | Y | Y | Y for RCT / only RCT | N | Y for RCT / N MA for NRSI | Y | Y | Y | Y | Y | High |
| Li-li Wang, et al. 2021 | Y | N | Y | PY | Y | Y | Y | Y | Y for RCT / N for NRSI | N | Y for RCT / N MA for NRSI | N | N | Y | N | Y | Critically Low |
| Maxwell Barnish & Barran 2020 | N | N | Y | PY | Y | Y | Y | Y | Y for RCT / Y for NRSI | Y | N for RCT / N MA for NRSI | N | Y | Y | N | Y | Critically Low |
| A. Berti, et al. 2020 | N | N | Y | PY | Y | Y | Y | N | Y for RCT / PY for NRSI | N | N MA | N MA | N | N | N MA | Y | Critically Low |
| Anna M. CarapellottiI, et al. 2020 | Y | N | Y | N | Y | N | Y | PY | Y for RCT / only RCT | N | Y for RCT / N MA for NRSI | Y | Y | Y | Y | Y | Critically Low |
| Kui Chen, et al. 2020 | Y | Y | Y | PY | Y | Y | Y | PY | Y for RCT / only RCT | N | Y for RCT / N MA for NRSI | Y | Y | Y | Y | Y | High |
| Heloisa de Almeida, et al. 2020 | Y | Y | Y | N | Y | Y | Y | Y | Y for RCT / only RCT | N | N for RCT / N MA for NRSI | Y | N | Y | N | Y | Critically Low |
| Ruben D. Hidalgo-Agudo, et al. 2020 | Y | N | Y | PY | Y | Y | PY | PY | Y for RCT / only RCT | N | Y for RCT / N MA for NRSI | Y | Y | Y | N | Y | Critically Low |
| Danique L. M. Radder, et al. 2020 | Y | N | Y | PY | N | Y | N | PY | PY for RCT / only RCT | N | Y for RCT / N MA for NRSI | Y | Y | Y | N | Y | Critically Low |
| Nadeesha Kalyani H. Haputhanthirigea, et al. 2019 | N | Y | Y | PY | Y | Y | Y | Y | Y for RCT / Y for NRSI | N | N for RCT / N MA for NRSI | Y | Y | N | N | N | Critically Low |
| Lijun Tang, et al. 2019 | N | N | N | PY | N | Y | Y | N | Y for RCT / PY for NRSI | N | Y for RCT/ N MA for NRSI | Y | Y | Y | N | Y | Critically Low |
| Qi Zhang, et al. 2019 | Y | N | Y | PY | Y | Y | Y | PY | Y for RCT / only RCT | N | Y for RCT / N MA for NRSI | Y | Y | Y | N | Y | Critically Low |

(*Continued*)

**Table 3.** (Continued)

| Author / Year | 1 | 2 | 3 | 4 | 5 | 6 | 7 | 8 | 9 | 10 | 11 | 12 | 13 | 14 | 15 | 16 | Overall Score |
|---|---|---|---|---|---|---|---|---|---|---|---|---|---|---|---|---|---|
| Marcela Delabary, et al. 2018 | Y | Y | Y | PY | Y | Y | Y | Y | PY for RCT / only RCT | N | Y for RCT / N MA for NRSI | Y | N | Y | N | Y | Critically Low |
| Adijatu Raheem & Casaca-Carreira 2018 | N | N | Y | N | N | N | Y | PY | Y for RCT / only RCT | N | N MA | N MA | N | Y | N MA | Y | Critically Low |
| Camila Monteiro Mazzarin, et al. 2017 | Y | N | Y | PY | Y | Y | Y | PY | PY for RCT / PY for NRSI | N | N for RCT / N MA for NRSI | N | N | N | N | N | Critically Low |
| Melanie E. Cusso, et al. 2016 | N | Y | Y | N | N | N | N | PY | Y for RCT / only RCT | N | N MA | N MA | Y | Y | N MA | Y | Critically Low |
| Jojo YanYan Kwok, et al. 2016 | Y | N | Y | PY | N | N | Y | PY | PY for RCT / PY for NRSI | N | N for RCT / N MA for NRSI | Y | Y | Y | Y | Y | Critically Low |
| Lorenna Pryscia C. Aguiar, et al. 2016 | N | N | Y | PY | Y | Y | N | PY | Y for RCT / Y for NRSI | N | N MA | N MA | Y | Y | N MA | Y | Critically Low |
| Luís A. A. Santos, et al. 2016 | Y | N | Y | N | Y | N | Y | PY | N for RCT / only RCT | N | N MA | N MA | N | N | N MA | Y | Critically Low |
| Priscila A. da Rocha, et al. 2015 | Y | N | Y | PY | Y | Y | PY | PY | Y for RCT / Y for NRSI | N | N for RCT / N MA for NRSI | N | N | N | N | Y | Critically Low |
| Désirée Lötzke, et al. 2015 | N | N | Y | PY | Y | Y | Y | Y | Y for RCT / Y for NRSI | N | Y for RCT / N MA for NRSI | N | N | N | N | Y | Critically Low |
| Ras lav Šumec, et al. 2015 | N | N | N | PY | N | N | PY | N | N for RCT / N for NRSI | N | N MA | N MA | N | N | N MA | Y | Critically Low |
| Joanne Shanahan, et al. 2015 | N | N | N | N | Y | N | PY | Y | Y for RCT / Y for NRSI | N | N for RCT / N MA for NRSI | N | Y | N | N | N | Critically Low |
| Danielle K. Murray, et al. 2014 | N | N | Y | PY | N | N | Y | PY | N for RCT / N for NRSI | N | N MA | N MA | N | N | N MA | Y | Critically Low |

(*Continued*)

**Table 3.** (Continued)

| Author / Year | 1 | 2 | 3 | 4 | 5 | 6 | 7 | 8 | 9 | 10 | 11 | 12 | 13 | 14 | 15 | 16 | Overall Score |
|---|---|---|---|---|---|---|---|---|---|---|---|---|---|---|---|---|---|
| Rosalind Mandelbaum & Lo 2014 | N | N | N | N | N | N | N | PY | N for RCT / N for NRSI | N | N MA | N MA | N | N | N MA | Y | Critically Low |
| Kathryn Sharp & Hewi 2014 | Y | N | Y | PY | Y | Y | Y | PY | Y for RCT / only RCT | N | N for RCT / N MA for NRSI | Y | Y | Y | N | N | Critically Low |
| Marie-Sophie Kiepe, et al. 2012 | N | N | Y | N | Y | Y | Y | PY | N for RCT / only RCT | N | N MA | N MA | N | N | N MA | N | Critically Low |
| M. J. de Dreu, et al. 2012 | N | N | Y | N | Y | Y | N | PY | Y for RCT / only RCT | N | Y for RCT / N MA for NRSI | Y | N | Y | N | Y | Critically Low |
| E. Valverde Guijarro & García 2012 | Y | N | Y | PY | N | N | N | PY | Y for RCT / N for NRSI | N | N MA | N MA | Y | N | N MA | Y | Critically Low |
| Claire L. Tomlinson, et al. (A) 2012 | Y | Y | Y | Y | Y | Y | Y | Y | Y for RCT / only RCT | N | Y for RCT / N MA for NRSI | Y | Y | Y | N | Y | Low |
| Claire L. Tomlinson, et al. (B) 2012 | Y | Y | Y | Y | Y | Y | Y | Y | Y for RCT / only RCT | N | Y for RCT / N MA for NRSI | Y | Y | Y | N | Y | Low |

*Note*. AMSTAR (Assessing the Methodological Quality of Systematic Reviews), Meta-analysis (MA), Y (Yes), N (No), PY (Partial Yes), Only RCT (Review included only RCTs), N MA (No Meta-Analysis performed).

*Items*. 1. Did the research questions and inclusion criteria for the review include the components of PICO? 2. Did the report of the review contain an explicit statement that the review methods were established prior to the conduct of the review and did the report justify any significant deviations from the protocol? 3. Did the review authors explain their selection of the study designs for inclusion in the review? 4. Did the review authors use a comprehensive literature search strategy? 5. Did the review authors perform study selection in duplicate? 6. Did the review authors perform data extraction in duplicate? 7. Did the review authors provide a list of excluded studies and justify the exclusions? 8. Did the review authors describe the included studies in adequate detail? 9. Did the review authors use a satisfactory technique for assessing the risk of bias (RoB) in individual studies that were included in the review? 10. Did the review authors report on the sources of funding for the studies included in the review? 11. If meta-analysis was performed did the review authors use appropriate methods for statistical combination of results? 12. If meta-analysis was performed, did the review authors assess the potential impact of RoB in individual studies on the results of the meta-analysis or other evidence synthesis? 13. Did the review authors account for RoB in individual studies when interpreting/ discussing the results of the review? 14. Did the review authors provide a satisfactory explanation for, and discussion of, any heterogeneity observed in the results of the review? 15. If they performed quantitative synthesis did the review authors carry out an adequate investigation of publication bias (small study bias) and discuss its likely impact on the results of the review? 16. Did the review authors report any potential sources of conflict of interest, including any funding they received for conducting the review?

To generate score: *https://amstar.ca/Amstar_Checklist.php*

**Table 4. Overall results of outcome measures based on The International Classification of Functioning, Disability and Health (ICF) core sets for Parkinson's disease.**

| Outcome | Outcome Measure | ICF Category | ICF Construct | Results from GRADE-Recommendations |
|---|---|---|---|---|
| Motor Symptoms Severity | Unified Parkinson's Disease Rating Scale, Motor Part (UPDRS III) | Body function (b) | Movement-related function (b) | ⊕⊕⊕○ **Moderate** (Dance↑ vs. Usual care) |
| Depressive Symptoms | Beck's Depression Inventory (BDI) | Body function (b) | Mental function (b) | ⊕⊕○○ **Low** (Dance↑ vs. Usual care) |
| Balance | Berg Balance Scale (BBS) | Body function (b) Activity (d) | Balance/falls (d) | ⊕⊕○○ **Low** (Dance↑ vs. Usual care) ⊕⊕⊕○ **Moderate** (Dance↑ vs. Multimodal Exercise) |
| Functional Mobility | Timed Up and Go Test (TUG) | Activity (d) | Balance/falls; gait; transfers (d) | ⊕⊕○○ **Low** (Dance↑ vs. Usual care) |
| Gait Distance | 6-minute walk test (6mWT) | Body function (b) Activity (d) | Aerobic capacity/ endurance; cardiovascular/ pulmonar status; fatigue (b) Gait (d) | ⊕⊕⊕○ **Moderate** (Dance ⇆ Usual care) |
| Quality of Life | The 39-item Parkinson's Disease Questionnaire (PDQ39) | Participation (d) | Quality of life (d) | ⊕⊕⊕○ **Moderate** (Dance ⇆ Usual care) |

*Note. Symbols:* '↑' experimental is superior to control, '⇆' experimental is equal to control.

For the six studies that included [83–88] dance-based intervention plus pharmacological usual care compared to pharmacological usual care alone, the intervention duration ranged from 8 to 12 weeks, twice a week for 60 to 90 minutes per session, except for one study [88] that participants danced only once a week. Individuals were between 64 and 70 years old, with mild to moderate PD symptom severity (based on the Hoehn & Yahr Scale). Two studies compared dance to multimodal exercise intervention, both in addition to pharmacological usual care [88, 89]. The intervention duration for these two studies ranged from 12 to 24 weeks, once a week, for 60 to 90 minutes each session. Individuals were 61 to 67 years old at mild to moderate stages of PD symptom severity (based on the Hoehn & Yahr Scale).

**Body function. Motor symptoms severity (UPDRS III).** Dance-based intervention reduced motor symptoms severity in comparison to pharmacological usual care alone (MD -2.26, 95% CI -3.29 to -1.22, p < 0.001; I2 0%; 4 RCTs) (Fig 2A). A sensitivity analysis showed that these values did not change significantly even when excluding the study by *Rios Romenets et al. (2015)* [85]. Finally, a linear regression test of funnel plot asymmetry (t = -0.44, df = 2, p-value = 0.70) confirmed that there is no bias publication for this outcome. The GRADE analysis showed that there is moderate-certainty evidence that dance plus pharmacological usual care is superior to pharmacological usual care alone for diminishing motor symptoms severity in people with PD (Table 5).

**Body function. Depressive symptoms (BDI).** Dance-based intervention plus pharmacological usual care diminished depressive symptoms in comparison to pharmacological usual care alone (MD -4.88, 95% CI -8.86 to -0.91, p = 0.01; I2 50%; 2 RCTs) (Fig 2B). The number of studies is too small to test for small study effects (publication bias). The GRADE analysis showed that there is low-certainty evidence that a dance-based intervention is superior to

**A. UPDRS III** (Motor Symptoms Severity)

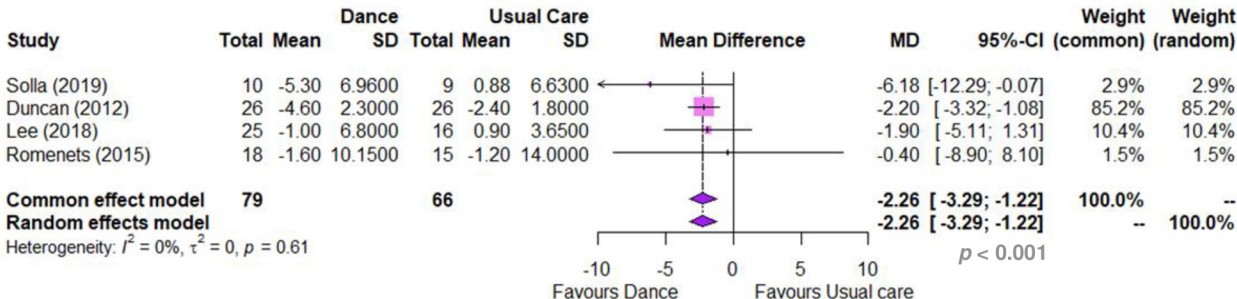

**B. BDI** (Depressive Symptoms)

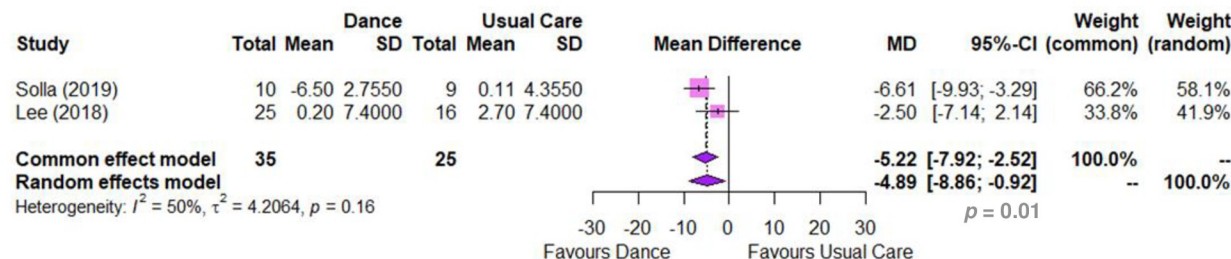

**Fig 2. Meta analyses.** (A) Dance compared to usual care regarding UPDRS III (Unified Parkinson's Disease Rating Scale, Motor Section); (B) Dance compared to usual care regarding BDI (Beck's Depression Inventory).

pharmacological usual care alone for diminishing depressive symptoms in people with PD (Table 5).

**Body function and activity. Balance (BBS).** Dance-based intervention plus pharmacological usual care improved balance capacity in comparison to pharmacological usual care alone (MD 3.24, 95% CI 0.64 to 5.83, p = 0.01; I2 68%; 4 RCTs) (Fig 3A). Linear regression test of funnel plot asymmetry (t = 0.75, df = 2, p-value = 0.5305) confirmed that there was no bias publication for this outcome. The GRADE analysis showed that there is low certainty of evidence that dance-based interventions are superior to pharmacological usual care alone in improving balance in people with PD (Table 5).

Dance-based intervention improved balance capacity in comparison to multimodal exercise intervention plus pharmacological usual care (MD 3.98, 95% CI 1.44 to 6.52, p = 0.002; I2 0%; 2 RCTs) (Fig 3B). The number of studies is too small to test for small study effects (publication bias). The GRADE analysis showed that there is moderate certainty of evidence that dance plus pharmacological usual care is superior to multimodal exercise intervention plus pharmacological usual care to improve balance capacity in people with PD (Table 5).

**Activity. Functional mobility (TUG).** Dance-based intervention plus pharmacological usual care improved functional mobility in comparison to pharmacological usual care alone (MD -1.14, 95% CI -1.84 to -0.43, p = 0.001; I2 22%; 4 RCTs) (Fig 4). When conducting a sensitivity analysis, values do not change significantly even when excluding the study by *Rios Romenets et al. (2015)* [85]. Moreover, a linear regression test of funnel plot asymmetry (t = 2.62, df = 2, p-value = 0.12) confirmed that there is no bias publication for this outcome. The GRADE analysis showed that there is low certainty of evidence that dance plus pharmacological usual care is superior to pharmacological usual care alone to improve functional mobility in people with PD (Table 5).

Table 5. Details of GRADE-recommendations.

**Question: Dance plus Pharmacological Usual Care compared to Pharmacological Usual Care for individuals with Parkinson's disease.**

| Certainty assessment | | | | | | | Nº of patients | | Effect | | Certainty | Importance |
|---|---|---|---|---|---|---|---|---|---|---|---|---|
| Nº of studies | Study design | Risk of bias | Inconsistency | Indirectness | Imprecision | Other considerations | Dance plus Pharmacological Usual Care | Pharmacological Usual Care | Relative (95% CI) | Absolute (95% CI) | | |
| **Motor Severity (follow-up: range 8 weeks to 12 weeks; assessed with: UPDRS III; Scale from: 0 to 132 (worse))** | | | | | | | | | | | | |
| 4 | randomised trials | not serious | not serious | not serious | serious[a] | none | 79 | 66 | - | MD **2.26 points lower** (3.29 lower to 1.22 lower) | ⊕⊕⊕◯ Moderate | CRITICAL |
| **Depression Symptoms (follow-up: range 8 weeks to 12 weeks; assessed with: BDI; Scale from: 0 to 63 (worse))** | | | | | | | | | | | | |
| 2 | randomised trials | not serious | serious[b] | not serious | serious[a] | none | 35 | 25 | - | MD **4.89 points lower** (8.86 lower to 0.92 lower) | ⊕⊕◯◯ Low | CRITICAL |
| **Balance (follow-up: range 8 weeks to 12 weeks; assessed with: BBS; Scale from: 0 to 56 (better))** | | | | | | | | | | | | |
| 4 | randomised trials | not serious | serious[c] | not serious | serious[a] | none | 80 | 54 | - | MD **3.24 points higher** (0.64 higher to 5.84 higher) | ⊕⊕◯◯ Low | CRITICAL |
| **Functional Mobility (follow-up: range 10 weeks to 12 weeks; assessed with: TUG; Scale from: Shorter time to Longer time (worse))** | | | | | | | | | | | | |
| 4 | randomised trials | not serious | serious[d] | not serious | serious[a] | none | 73 | 53 | - | MD **1.44 seconds fewer** (1.84 fewer to 0.44 fewer) | ⊕⊕◯◯ Low | CRITICAL |
| **Gait distance (follow-up: range 10 weeks to 12 weeks; assessed with: 6MWT; Scale from: Shorter time to Longer time (better))** | | | | | | | | | | | | |
| 2 | randomised trials | not serious | not serious | not serious | serious[a] | none | 56 | 41 | - | MD **22.91 meters higher** (12.61 lower to 58.43 higher) | ⊕⊕⊕◯ Moderate | CRITICAL |
| **Quality of Life (follow-up: range 10 weeks to 12 weeks; assessed with: PDQ39; Scale from: 0 to 100 (worse))** | | | | | | | | | | | | |
| 2 | randomised trials | not serious | not serious | not serious | serious[a] | none | 48 | 30 | - | MD **0.69 points higher** (5.32 lower to 6.71 higher) | ⊕⊕⊕◯ Moderate | CRITICAL |

*Question:* **Dance plus Pharmacological Usual Care** compared to **Multimodal Exercise** for individuals with Parkinson's disease

(*Continued*)

**Table 5.** (Continued)

**Question: Dance plus Pharmacological Usual Care compared to Pharmacological Usual Care for individuals with Parkinson's disease.**

| Nº of studies | Study design | Risk of bias | Inconsistency | Indirectness | Imprecision | Other considerations | Dance plus Pharmacological Usual Care | Multimodal Exercise / Pharmacological Usual Care | Relative (95% CI) | Absolute (95% CI) | Certainty | Importance |
|---|---|---|---|---|---|---|---|---|---|---|---|---|
| | | | Certainty assessment | | | | Nº of patients | | Effect | | | |
| **Balance (follow-up: range 12 weeks to 24 weeks; assessed with: BBS; Scale from: 0 to 56 (better))** | | | | | | | | | | | | |
| 2 | randomised trials | not serious | not serious | not serious | serious[a] | none | 27 | 29 | - | MD **3.98 score higher** (1.44 higher to 6.52 higher) | ⊕⊕⊕◯ Moderate | CRITICAL |

**CI:** confidence interval; **MD:** mean difference

Explanations

a. The total number of participants in this comparison is lower than the Optimal Information Size.

b. Heterogeneity: I2 50%, p = 0.16

c. Heterogeneity: I2 68%, p = 0.03

d. Kunkel (2017) trial results in terms of functional mobility are inconsistent with the remaining trials.

**CI:** confidence interval; **MD:** mean difference

Explanations

a. The total number of participants is lower than the Optimal Information Size.

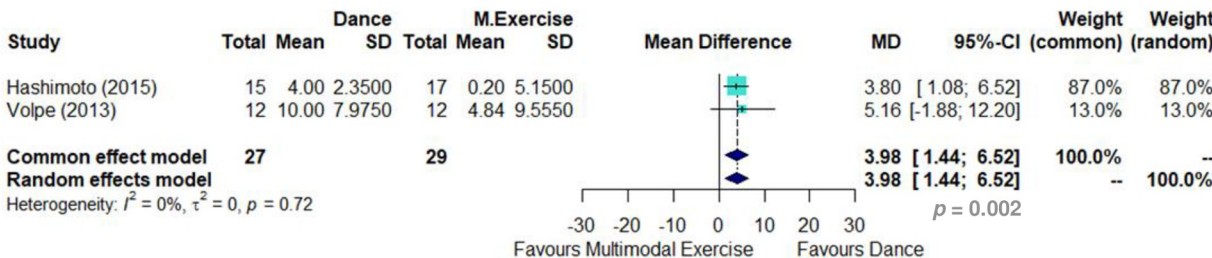

**A. BBS** (Balance)

**B. BBS** (Balance)

**Fig 3. Meta analyses.** (A) Dance compared to usual care regarding BBS (Berg Balance Scale); (B) Dance compared to multimodal exercise regarding BBS (Berg Balance Scale).

**Body function and activity. Gait distance (6MWT).** Dance-based intervention plus pharmacological usual care does not provide benefits over and above pharmacological usual care alone (MD 22.91, 95% CI -12.61 to 58.43, p = 0.20; I2 0%; 2 RCTs) (Fig 5A). When conducting a sensitivity analysis, we excluded the study of Solla [86] from the meta-analysis because the delta difference was much higher (MD = 239) than other studies, with a higher heterogeneity (I2 89%). Overall, the number of studies is too small to test for small study effects (publication bias). The GRADE analysis showed that there is moderate-certainty evidence that dance plus pharmacological usual care is similar to pharmacological usual care alone to improve functional mobility in people with PD (Table 5).

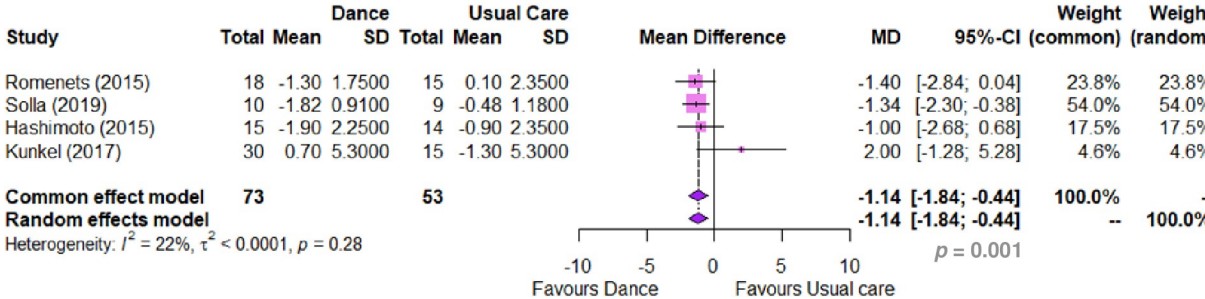

**TUG** (Functional Mobility)

**Fig 4. Meta analysis.** Dance compared to usual care regarding TUG (Timed Up and Go).

**A.  6MWT** (Gait Distance)

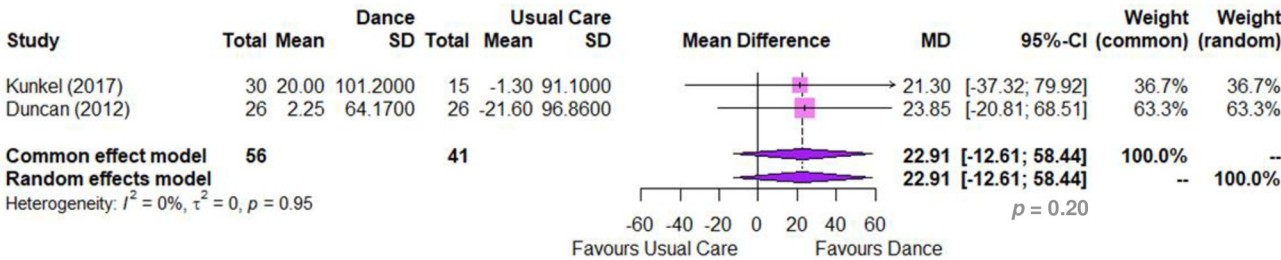

**B.   PDQ39** (Quality of Life)

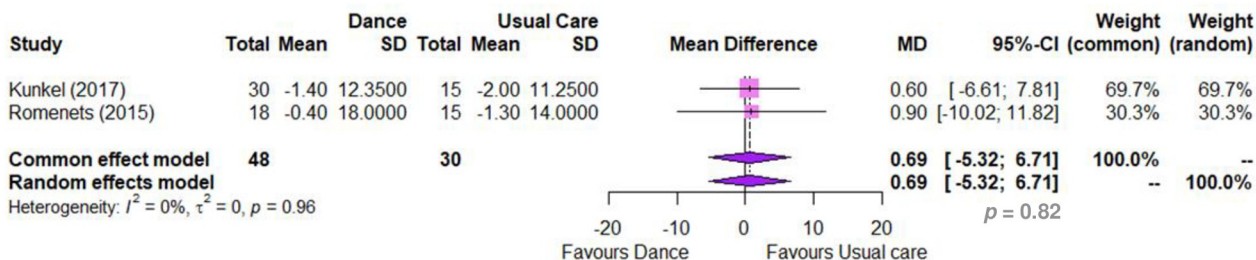

**Fig 5. Meta analyses.** (A) Dance compared to usual care regarding 6MWT (Six Minute Walking Test); (B) Dance compared to usual care regarding PDQ-39 (Parkinson's Disease Questionnaire).

**Participation. Quality of life (PDQ39).** Dance-based intervention plus pharmacological usual care provide similar benefits compared to pharmacological usual care alone for quality of life (MD 0.69, 95% CI -5.32 to 6.70, p = 0.82; I2 0%; 2 RCTs) (Fig 5B). We did not include Lee (2018) [84] in this meta-analysis because the authors used another outcome measurement for quality of life (PDQL) [90]. When conducting a sensitivity analysis, values do not change significantly, even when excluding the study by *Rios Romenets et al. (2015)* [85]. The number of studies is too small to test for small study effects (publication bias). The GRADE analysis showed that there is moderate-certainty evidence that dance plus pharmacological usual care is similar to pharmacological usual care alone for quality of life in people with PD (Table 5).

## Overall results of outcome measurements based on ICF

Table 4 presents outcome measures matched with ICF guidelines and core sets for PD [2–4]. The three components of ICF with their respective constructs/categories were shown accordingly. Overall results based on GRADE Recommendations were also detailed (details of GRADE recommendations are available in Table 5).

## Discussion

As far as we know, this is the first quality assessment study and umbrella review of systematic reviews and about dance-based intervention effectiveness for individuals with PD. From 48 reviews included, only six were considered high quality based on the AMSTAR 2 tool. We also updated and extended the understanding of dance as an adjunct rehabilitation to pharmacological usual care on ICF outcomes effectiveness in people with mild to moderate PD. We found moderate-certainty evidence to recommend dance-based intervention for people with

PD to ameliorate ICF body functions and activity domains, such as motor symptoms severity and balance, compared to pharmacological usual care alone or multimodal exercises, respectively.

Overall, the quality of the included systematic reviews on the effectiveness of dance for people with PD was rated critically low to low (i.e. 85%). Previous methodological survey studies on the effectiveness of exercise also reported critically low confidence in the results of the majority of systematic reviews (i.e. 74%), for example in the context of lower back pain [91]. Another systematic review [45] also assessed the quality of studies included on dance interventions for people with neurodegenerative conditions such as mild cognitive impairment, Alzheimer's disease, and PD. According to AMSTAR and PEDro criteria, reviews and RCTs were moderate to low quality [45]. However, the authors applied the first version of *AMSTAR tool 1* (11 questions) for only 16 reviews. In our study, we used the newest version of *AMSTAR tool 2* (16 questions) and included 55 reviews in the field. Surprisingly, we found several reviews and a low number of low risks of biased randomized clinical trials. Most of the different results are due to the inclusion of clinical trials with a high risk of bias [50, 53], non-RCT [56] and combining music therapy, aquatic therapy [50] or other exercises [36] into the dance-based meta-analyses. We found that the only dance-specific review with high quality and overall confidence is Ismail (2021) [25].

The overall confidence for the majority of included reviews was mostly downgraded due to study limitations on AMSTAR critical domains. More than the overall score, reviews might be interpreted considering the impact of each individual item of AMSTAR [92]. In our results, more than two-thirds of reviews did not report a protocol registration (e.g., PROSPERO or COCHRANE) and did not appraise all relevant aspects of the literature search. The low quality found was also a consequence of not considering the impact of risk of bias on discussing the results. The AMSTAR 2 tool does not look at the quality of clinical studies included in reviews but how well the review provided a transparent discussion about the potential risk of bias and heterogeneity, which is not covered if the review simply assessed the risk of bias. Also, it is not recommended to cluster RCTs and non-RCTs in the same meta-analyses [39, 93]. Among the reviews with meta-analyses, many did not assess potential publication bias by performing a sensitivity analysis (e.g., funnel plot asymmetry perhaps due to the lack of a sufficient number of studies to include.

Some systematic reviews with meta-analysis included RCTs that were, in fact, non-randomized or had very small sample size in meta-analysis. As an attempt to update and compile low-risk-of-bias RCTs regarding the effects of dance-based interventions on the PD population, we conducted pairwise meta-analyses of low-risk-of-bias RCTs focusing on outcomes that match the ICF domains. We found moderate certainty of evidence for dance plus pharmacological usual care improving body functions, such as motor symptoms severity (UPDRS-III; MD = 2.26), compared to pharmacological usual care alone. However, the mean difference of 2.26 points may not reach the minimal clinically important difference for UPDRS-III in people's daily lives [94]. Our results support existing American guidelines from 2022 [12] but not the European guidelines [2] from 2014, possibly because the latter do not consider more recent evidence. We also showed moderate-certainty evidence that dance plus pharmacological usual care is superior to multimodal exercise plus pharmacological usual care to improve body function and activity, such as balance (BBS; MD = 3.98). Dance typically requires a certain level of balance, for example to step in multiple directions and these balance requirements can be designed to be more challenging if performed in choreographies or dance improvisation [2, 51]. This helps to explain why we found that dance is better at improving balance capacity than multimodal exercise programs, including balance training. In conclusion, our results showed that dance associated with pharmacological usual care reduced depressive symptoms

(BDI; MD = 4.89), and improved balance (BBS; MD = 3.24) and functional mobility (TUG; MD = -1.44 sec) when compared to pharmacological usual care alone.

The BBS tool evaluates balance by looking at activities, such as reaching, turning, and unipedal stance. The ICF checklist from WHO evaluates the outcome balance as a body function, although European and American PD recommendations[1,2,86] considered it an activity domain. However, this conceptual conflict over balance classification can lead to confusion, and recommendations should start from the same assumption to minimize these differences. Finally, our meta-analyses found that dance plus pharmacological usual care is similar to pharmacological usual care alone for improving gait distance (6MWT; MD = 22.91) and quality of life (PDQ39; MD = 0.69). Although dance can increase cardiovascular resistance over practice, this might not be sufficient to improve gait distance during six minutes. In terms of quality of life, the PDQ-39 tool includes the frequency of difficulties the person reports for mobility, daily basic activities, emotional well-being, stigma, social support, cognition, verbal communication, and body discomfort. The extent to which dance could underpin several aspects of quality of life is complex to measure and it might require more than twelve weeks of intervention to achieve a clinical effect plausible.

## Limitations

Overall quality scores for each systematic review included in our umbrella study must be interpreted cautiously, especially when comparing reviews with and without meta-analysis. Most of our meta-analysis results presented 0% heterogeneity, except for depressive symptoms and balance compared to pharmacological usual care that showed substantial heterogeneity (50%-70%) and so low certainty of evidence. Moreover, variations in healthcare systems, medications, and cultural differences in dance practice and traditions across regions of the globe might influence the results and overall well-being in individuals diagnosed with PD. Each country seems to use a traditional form of dance style, in keeping with local tradition and culture. Our study does not recommend a specific dance style for the general PD population; on the contrary, we recommend using the most popular and familiar dance style within a specific population.

## Recommendations

In our meta-analysis results, most of the participants were older adults (over 60 years) at mild to moderate stages of PD symptom severity [95]. We recommend a dance intervention dose twice weekly, 60 minutes each session, for at least 12 weeks. However, RCTs included did not describe the intervention protocol in detail, as suggested by Template for Intervention Description and Replication (TIDieR) [96]. Trials rarely reported how the program advanced, such as frequency, intensity, time, type, volume, and progression (FITT-VP) [97]. In some studies, it is unclear whether the intervention involved solo or group dancing, although typically dance is practiced and performed in groups. Finally, it was unclear for many studies if the dance protocol included in clinical trials was or could be adapted to sitting on a chair, standing with support or both. We encourage future clinical trials to provide more detail on the specifics of the dance intervention protocol.

We recommend conducting more RCTs that compare dance with other modalities of therapeutic exercise in people with PD to establish consistent quality guidelines. Future RCTs should adopt standardized templates, such as TIDieR [96], to enhance comparability of studies and interventions, thereby facilitating translation into clinical practice. We strongly recommend systematic reviews to differentiate between non-randomized controlled trials (NRCT) and RCTs in meta-analyses for different outcomes and to select unbiased studies. Additionally,

future reviews should adhere to PRISMA, Cochrane, and AMSTAR guidelines and should pre-register protocols. The quality and confidence of reviews concerning other active forms of therapeutic or physical exercise, apart from dance, for people with PD remain uncertain. We also encourage future studies to align outcome measurements with the International Classification of Functioning (ICF) components relevant to the PD population.

## Conclusions

We captured the most relevant literature and provided directions for recommendations of seven reviews with high overall confidence. This understanding is critical to generate unbiased estimates of treatment effects for decision-making and alerting clinicians to base their practical conduct only on high-quality reviews. Additionally, we integrated ICF components by updating meta-analyses on the effectiveness of dance-based interventions for the PD population. Dance-based interventions combined with pharmacological usual care are recommended to improve body functions and activities as categorized by the WHO International Classification of Functioning, Disability and Health, by reducing motor symptom severity and depression, and by improving balance and functional mobility. Elucidate these findings to strengthen the use of dance for people with PD and its potential implications on patient-reported outcomes.

## Supporting information

**S1 Checklist. PRISMA 2020 checklist.**
(DOCX)

**S1 File. Search strategies.**
(DOCX)

**S1 Table. Checklists of included reviews.**
(DOCX)

**S1 Data. Research results and data extracted from the primary research sources.**
(XLSX)

## Author Contributions

**Conceptualization:** Camila Pinto.

**Data curation:** Camila Pinto, Rafaela Simon Myra, Francisca Pereira.

**Formal analysis:** Camila Pinto, Alexandre Severo do Pinho, Francisca Pereira.

**Investigation:** Camila Pinto, Rafaela Simon Myra, Guido Orgs, Aline Souza Pagnussat.

**Methodology:** Camila Pinto, Rafaela Simon Myra, Alexandre Severo do Pinho, Francisca Pereira.

**Project administration:** Camila Pinto, Rafaela Simon Myra, Aline Souza Pagnussat.

**Resources:** Camila Pinto, Guido Orgs, Aline Souza Pagnussat.

**Supervision:** Guido Orgs, Aline Souza Pagnussat.

**Visualization:** Camila Pinto, Alexandre Severo do Pinho, Aline Souza Pagnussat.

**Writing – original draft:** Camila Pinto.

**Writing – review & editing:** Camila Pinto, Rafaela Simon Myra, Alexandre Severo do Pinho, Francisca Pereira, Guido Orgs, Aline Souza Pagnussat.

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
