## [Decision Letter · Decision Letter 0]

22 Apr 2024

PONE-D-23-44080Quality Assessment and Umbrella Review of Systematic Reviews about Dance for People with Parkinson's DiseasePLOS ONE

Dear Dr. Pinto,

Thank you for submitting your manuscript to PLOS ONE. After careful consideration, we feel that it has merit but does not fully meet PLOS ONE’s publication criteria as it currently stands. Therefore, we invite you to submit a revised version of the manuscript that addresses the points raised during the review process.

Please read the reviewers' comments carefully and respond to all of them in detail and submit your revised manuscript by Jun 06 2024 11:59PM. If you will need more time than this to complete your revisions, please reply to this message or contact the journal office at plosone@plos.org. Please include the following items when submitting your revised manuscript:A rebuttal letter that responds to each point raised by the academic editor and reviewer(s). You should upload this letter as a separate file labeled 'Response to Reviewers'.A marked-up copy of your manuscript that highlights changes made to the original version. You should upload this as a separate file labeled 'Revised Manuscript with Track Changes'.An unmarked version of your revised paper without tracked changes. You should upload this as a separate file labeled 'Manuscript'.If applicable, we recommend that you deposit your laboratory protocols in protocols.io to enhance the reproducibility of your results. Protocols.io assigns your protocol its own identifier (DOI) so that it can be cited independently in the future. For instructions see: https://journals.plos.org/plosone/s/submission-guidelines#loc-laboratory-protocols. Additionally, PLOS ONE offers an option for publishing peer-reviewed Lab Protocol articles, which describe protocols hosted on protocols.io. Read more information on sharing protocols at https://plos.org/protocols?utm_medium=editorial-email&utm_source=authorletters&utm_campaign=protocols.

We look forward to receiving your revised manuscript.

Kind regards,

Rainbow T. H. Ho

Academic Editor

PLOS ONE

2. Please upload a copy of Figure 3a and 3b, to which you refer in your text on page 12. If the figure is no longer to be included as part of the submission please remove all reference to it within the text.

3. Please include your tables as part of your main manuscript and remove the individual files. Please note that supplementary tables (should remain/ be uploaded) as separate ""supporting information"" files".

Reviewers' comments:

Reviewer's Responses to Questions

**Comments to the Author**

1. Is the manuscript technically sound, and do the data support the conclusions?

Reviewer #1: Yes

Reviewer #2: Yes

2. Has the statistical analysis been performed appropriately and rigorously? 

Reviewer #1: I Don't Know

Reviewer #2: Yes

3. Have the authors made all data underlying the findings in their manuscript fully available?

Reviewer #1: Yes

Reviewer #2: Yes

4. Is the manuscript presented in an intelligible fashion and written in standard English?

Reviewer #1: Yes

Reviewer #2: No

5. Review Comments to the Author

Reviewer #1: This paper provides a comprehensive, up to date review of the effectiveness of Dance interventions for people with Parkinsons disease. The paper is clearly organised, for the most part it reads well.

Major Concerns

The review argues that there are two different guidelines (American and European) with recommendations that contradict one another. However, the European guidelines came out in 2014 and the American guidelines are more recent as of 2022. Contradictions may be explained by one being more recent than the other with more up-to-date evidence, particularly since PD is becoming more prevalent. If this is the case, is it not really a contradiction, and it is unclear why this umbrella review is needed. It is recommended that the authors provide a case for what this review adds to the 2022 guidelines.

Furthermore, the guidelines are from different places with different healthcare regulations, medications etc – could be hard to compare? Is it possible they are based on evidence from different continents?

Minor Concerns

Introduction

- Maybe should have the guidelines of the ICF in the paper somewhere. (table 3?)

- 71, what symptoms does it not cover/which side effects become more severe? Are these symptoms/side effects ones that dance can help with?

- From line 80, feels contradictory. Is there recommendations or not?

- Line 85, although they are contradictory does it matter if one is European and the other is American – different healthcare systems, different medications etc.

- Line 110, now talking about social participation too – consider introducing earlier.

Objectives

- Objectives slightly differ to the ones in the abstract - framework not mentioned, please ensure consistency.

Methods

- What is a methodological survey, is it AMSTAR tool? Why has it been separated from the systematic review of reviews, should it be part of that?

Search Methods

- Needs clarity around experts consulted, are they medical professionals or something else?

- Clarify the dates when databases were searched, being more specific, not just reviewed up until April 2023.

- Line 157, PICO? Is PICOT meant here?

Data Collection and Selection

- How was the consensus reached?

Data Extraction and Management

- Consensus?

- 175, what is NRSI?

- Line 190: What did the pilot trial consist of?

- Figure 1: Why were 129 records excluded after screening?

Results

- Results: Para starting line 280 – do you mean that 5 studies had 2x sessions per week and one study 1x session per week? If so this could be made clearer.

- Results: Should the measures not be in the methods rather than results? E.g. statements such as “Motor symptoms severity was evaluated by means Unified Parkinson’s Disease Rating Scale, 293 Motor Section (UPDRS III).”

- 297, educational paper material, it would be good to clarify what this means.

- Repetitive language, could just explain something once e.g. excluding Romenets and explaining educational paper material.

Discussion

- 379, methodological survey, please clarify what this is.

- Discussion: Line 399, this sentence does not seem to follow from the previous point, “In fact, some RCTs included were non-randomized or provided a small sample size.”

Limitations

- Lines 448 and 455, recommendations in limitations, could have own section.

- Line 411 – the mean change in what? Is this going back to your own results? Perhaps make this clearer as the last sentence was about the guidelines.

Reviewer #2: This umbrella review can potentially contribute to the field. However, more elaboration or clarification on the following aspects is needed.

Introduction

1. The entire review was written in a very technical sense. To enhance the value and significance of this review, the rationale behind choosing “dance and Parkinson’s disease” as the topic can be further elaborated. For example, the definitions of Dance and Dance-based intervention. How did other systematic reviews conceptualize them? Did they consider Dance and Dance-based intervention as a kind of physical exercise, recreational program, or psychosocial intervention? What is/are the characteristic(s) of Dance as an art form? What is/are the difference(s) between Dance and other forms of physical exercise?

For example, in Page 4, Line 79-80, the authors defined Dance as “an art-based exercise guided by music that works on similar physical elements when compared to other forms of multimodal training for PD”. I would be curious if music is the more important element for the rehabilitation of Parkinson’s disease.

By elaborating these aspects and linking them up with the results of this review, the conclusion may be more sound.

Methods

2. Since this review has no language restriction, just curious if the authors encountered any research studies that were written in other languages they were not familiar with. If yes, how did they handle this issue?

3. Page 7, Line 175, the sentence “The frequency of each RCT was also computed.”: Does the word “frequency” refer to “intervention dosage” or “numbers of intervention sessions”?

Result

4. Page 10, Line 251-260: Similar information has been reported in the Methods section. By refining the contents, the manuscript could be more concise.

5. Page 11, Line 260: It should be “34” reviews with meta-analysis.

6. For the section “Umbrella Review”, some of the contents should be put in Methods section.

Discussion

7. Page 17, Line 433: The meaning of the phrase “more than twelve to ten weeks of intervention” was a bit ambiguous.

Limitations

8. Page 17, Line 445: Does the term MA refer to meta-analysis?

Last but not least, professional English editing is needed for this manuscript.

6. PLOS authors have the option to publish the peer review history of their article (what does this mean?). If published, this will include your full peer review and any attached files.

Reviewer #1: No

Reviewer #2: No

---

## [Author Response · Author response to Decision Letter 0]

22 Jul 2024

1. When submitting your revision, we need you to address these additional requirements. Please ensure that your manuscript meets PLOS ONE's style requirements, including those for file naming. The PLOS ONE style templates can be found at 

Response: Thank you for your guidance regarding the templates. We have confirmed that the revised manuscript adheres to PLOS ONE's style requirements, including correct file naming conventions. Please do not hesitate to contact us if the manuscript requires any further adjustments.

2. Please upload a copy of Figure 3a and 3b, to which you refer in your text on page 12. If the figure is no longer to be included as part of the submission please remove all reference to it within the text.

Response: Thank you for pointing out this typo. Figures 3a and 3b are now corrected to Figures 2a and 2b.

3. Please include your tables as part of your main manuscript and remove the individual files. Please note that supplementary tables (should remain/ be uploaded) as separate ""supporting information"" files".

Response: The tables have been integrated into the main document, apart from supporting files.

Response: We have carefully reviewed the Supporting Information files and ensured that our manuscript complies with these requirements.

5. Please review your reference list to ensure that it is complete and correct. If you have cited papers that have been retracted, please include the rationale for doing so in the manuscript text or remove these references and replace them with relevant current references. Any changes to the reference list should be mentioned in the rebuttal letter that accompanies your revised manuscript. If you need to cite a retracted article, indicate the article’s retracted status in the References list and also include a citation and full reference for the retraction notice.

Response: We reviewed all references included and we noticed that one of them (Hao Z. et al 2022) was retracted and corrected as detailed below:

Hao, Zikang, Xiaodan Zhang, and Ping Chen. 2023. "Correction: Hao et al. Effects of Ten Different Exercise Interventions on Motor Function in Parkinson’s Disease Patients—A Network Meta-Analysis of Randomized Controlled Trials. Brain Sci. 2022, 12, 698" Brain Sciences 13, no. 6: 885. https://doi.org/10.3390/brainsci13060885

We now cite this correct version in our manuscript. Importantly, this correction did not interfere with our quality assessment evaluation by the AMSTAR tool. 

Another review included in our analysis was recently updated by Cochrane collaboration. Ernst M. et al 2023 is now updated and cited as Ernst M. et al 2024 in the whole document. Our quality assessment evaluation by the AMSTAR tool did not change.

When comparing the old version to the new, we noticed only one difference, as detailed below and corrected in our tables:

- RCTs referenced in 2023 → 13 studies (Duncan 2012; Hackney 2007; Hackney 2009; Kunkel 2017; Michels 2018; Pohl 2013; Pohl 2020; Poier 2019; Rios Romenets 2015; Shanahan 2017; Solla 2019; Terrens 2020; Volpe 2013) 

- RCTs referenced in 2024 → 12 studies (Duncan 2012; Hackney 2007; Hackney 2009; Kunkel 2017; Michels 2018; Pohl 2013; Pohl 2020; Poier 2019; Rios Romenets 2015; Shanahan 2017; Solla 2019; Volpe 2013). 

In the new version, authors excluded the reference of Terrens 2020 because this study is not about dance intervention. RCTs included in each meta-analysis for each outcome did not change in the old and new version. We believe authors referenced Terrens 2020 as a typo.

Response to Reviewer 1

This paper provides a comprehensive, up to date review of the effectiveness of Dance interventions for people with Parkinsons disease. The paper is clearly organised, for the most part it reads well.

Response: We thank the reviewer for dedicating time and effort to helping us to improve this work. The reviewer pointed out important questions that helped us to improve the manuscript. The work the reviewer spent on our paper was extremely valuable. All the comments were relevant, and we did our best to answer all of them. The responses to each comment or suggestion are detailed below. In addition to addressing the reviewer's comments, we conducted an updated search independently to ensure we did not overlook any recent systematic reviews. This effort included updating the database search through June 2024. Identifying seven additional revisions necessitated thorough data analysis, which contributed to the extended timeline.

Major Concerns

The review argues that there are two different guidelines (American and European) with recommendations that contradict one another. However, the European guidelines came out in 2014 and the American guidelines are more recent as of 2022. Contradictions may be explained by one being more recent than the other with more up-to-date evidence, particularly since PD is becoming more prevalent. If this is the case, is it not really a contradiction, and it is unclear why this umbrella review is needed. It is recommended that the authors provide a case for what this review adds to the 2022 guidelines. Furthermore, the guidelines are from different places with different healthcare regulations, medications etc – could be hard to compare? Is it possible they are based on evidence from different continents?

Response: The reviewer is completely right. The contradiction between guidelines might be due to the emerging evidence since 2014. We changed this paragraph since it is not our aim to analyze quality and compare guidelines, but yes cite recommendations and analyze systematic reviews. 

Pages 4-5, lines 82-92: 

“European [1] and American [11,12] Guidelines for physiotherapy in PD include dance as a therapeutic, multimodal and community-based modality recommended for people with PD. The European Physiotherapy Guideline for PD (2014) [1] recommends dance with a moderate level of certainty of slight benefit on functional mobility, but no recommendations to ameliorate quality of life and motor symptoms severity [1]. More recently, the American Guideline (2022) [11] provides strong recommendations for community-based exercise to improve functional mobility, and quality of life and reduce motor symptoms severity, but no recommendation to improve balance. However, the Guideline [11] refers to community-based exercise modalities, such as dance, yoga, pilates, multimodal training, and others. Thus, it is not clear whether dance could improve outcomes cited above. Future research should focus on a target modality due the range of variability between dance and other community-based modalities [11].”

Minor Concerns

Introduction

- Maybe should have the guidelines of the ICF in the paper somewhere. (table 3?)

Response: We acknowledged the necessity of clarifying the ICF guidelines and addressed this by providing detailed information and references in the first paragraph of the introduction section as well as in Table 3. We believe that references and detailed information provided will help ensure that readers will quickly understand and locate the guidelines recommendations.

- 71, what symptoms does it not cover/which side effects become more severe? Are these symptoms/side effects ones that dance can help with?

Response: Common side effects include dyskinesias, sleep problems and anxiety. Exercise has been documented as an important strategy, potentially counteracting some of the anxiety-inducing effects of levodopa treatment. We value your questions, and we have now highlighted this information:

Page 4, lines 72-77: 

“The most known treatment with pharmacological usual care (e.g. levodopa) does not cover all PD symptoms, and side effects, such as dyskinesias and anxiety, can become more severe with chronic administration.”

“Exercise would have a potential to alleviate anxiety associated with levodopa treatment in individuals with PD.” 

- From line 80, feels contradictory. Is there recommendations or not?

Response: We agree with the reviewer. We changed the paragraph as detailed below.

Pages 4-5, lines 88-95: 

“The European Physiotherapy Guideline for PD (2014) [2] recommends dance for improving functional mobility, but not as an intervention to support quality of life or motor symptoms severity [2]. More recently, the American Guideline (2022) [12] provides strong recommendations for community-based exercise to improve functional mobility and quality of life and reduce motor symptoms severity. It includes activities such as dance, yoga, Pilates, but does not provide any specific evidence that dance can improve PD symptoms or quality of life. Future research should focus on a target modality due to the range of variability between dance and other community-based modalities [12]. "

- Line 85, although they are contradictory does it matter if one is European and the other is American – different healthcare systems, different medications etc.

Response: The reviewer is right. Variations in healthcare systems and medications between Europeans and Americans can affect their lives in several ways. These disparities may influence health outcomes and overall well-being. We discussed this topic further in the discussion section (lines 496-502).

- Line 110, now talking about social participation too – consider introducing earlier.

Response: Thank you for pointing that out. We delete the word “social” due to its double meaning. We mean participation as a domain of the International Classification of Functioning, Disability, and Health (ICF) from WHO, which classifies quality of life as a participation domain. We now explain that in the first paragraph of the introduction section as previously requested. 

Objectives

- Objectives slightly differ to the ones in the abstract - framework not mentioned, please ensure consistency.

Response: We have aligned the objective statements between the abstract and introduction sections and removed the word "framework" throughout the manuscript.

Methods

- What is a methodological survey, is it AMSTAR tool? Why has it been separated from the systematic review of reviews, should it be part of that?

Response: Our paper provides an umbrella review of existing reviews on the efficacy of dance-based interventions for PD). We intend to conduct a methodological survey of systematic reviews, recognizing that this approach may be misconstrued as part of the systematic review of reviews. The words 'Methodological survey' were excluded from the sentence on page 6 (lines 137-138).

Search Methods

- Needs clarity around experts consulted, are they medical professionals or something else?

Response: We consulted experts - health professionals with backgrounds in dance and PD - to ensure we didn't miss any additional reviews (page 7 – lines 164-166). According to AMSTAR tool, additional sources to complement searches include consulting published reviews, specialized registers, or experts in the relevant field, as well as reviewing the reference lists of identified studies. Consulting experts is not mandatory; therefore, this information can be omitted if the reviewer deems it unnecessary.

- Clarify the dates when databases were searched, being more specific, not just reviewed up until April 2023.

Response: We now detailed the starting point of searching and updated the search in June 2024, as detailed below (page 7, lines 160-163):

“We started the search on March 2023 and consulted the databases MEDLINE, PUBMED, Embase, Scopus, CENTRAL (Cochrane Library), CINAHL, PEDro, SPORTDiscus, APA PsycNet (APA PsycINFO), LILACS, SciELO, and AMED. We finished the selection in May 2023 and updated the search in June 2024.”

- Line 157, PICO? Is PICOT meant here?

Response: Yes, we corrected to “PICOT”.

Data Collection and Selection

- How was the consensus reached?

Response: We apologize for this mistake. This part of the manuscript is not for referring to results but yes to detail how was the process of data selection done. We modified this sentence, as detailed below:

Page 8, lines 175-176:

“Then, two reviewers (CP and RS) independently read full articles eligible, any disagreements were resolved through discussion with a third (ASP) reviewer when necessary.”

Data Extraction and Management

- Consensus?

Response: As explained above, this part of the manuscript is not for referring to results about consensus but yes to detail how was the process of data extraction done. We modified this sentence, as detailed below:

Page 8, lines 178-179:

“Two authors performed the data extraction independently (CP and RS or FP) and then cross-checked each other’s work.”

- 175, what is NRSI?

Response: This is a typo, we corrected it to “NRCT”.

- Line 190: What did the pilot trial consist of?

Response: Thank you for pointing that out, we detail this information on page 9, lines 207-213:

“Reviewers learned how to use AMSTAR-2 by referred literature and carried out a training trial before starting the evaluation of articles included in this paper. We used a previous quality assessment paper already published to learn and train our evaluation and discuss disagreements with an expert in the field. This training trial took around one month. Finally, each reviewer (CP and RSM) independently assessed the included articles and compared the results. Discrepancies were handled in a consensus dialogue in a group and remaining disagreements were resolved by a third author (ASP).”

- Figure 1: Why were 129 records excluded after screening?

Response: 129 studies were excluded after the title and abstract assessment, according to eligibility criteria. We detail this information in Figure 1.

Results

- Results: Para starting line 280 – do you mean that 5 studies had 2x sessions per week and one study 1x session per week? If so this could be made clearer.

Response: We corrected the sentence to improve clarity (page 26, lines 336-339).

“For the six studies that included [85-89,91] dance-based intervention plus pharmacological usual care compared to pharmacological usual care alone, the intervention duration ranged from 8 to 12 weeks, twice a week for 60 to 90 minutes per session, except for one study [89] that participants danced only once a week.”

- Results: Should the measures not be in the methods rather than results? E.g. statements such as “Motor symptoms severity was evaluated by means Unified Parkinson’s Disease Rating Scale, 293 Motor Section (UPDRS III).”

Response: We included this information about the tests exclusively in the methods section and added a sentence indicating that all outcomes were analyzed using the same tool (page 9, lines 219-227).

“Outcomes related to the International Classification of Functioning, such as motor symptom severity (body function), depressive symptoms (body function), balance (body function and activity), functional mobility (activity), gait distance (body function and activity), and quality of life (participation), were assessed. Motor symptom severity was evaluated using the Unified Parkinson’s Disease Rating Scale, Motor Section (UPDRS III). Depressive symptoms were assessed using Beck’s Depression Inventory (BDI). Balance was measured using the Berg Balance Scale (BBS). Functional mobility was evaluated with the Timed Up and Go (TUG) test. Gait distance was assessed using the Six Minute Walking Test (6MWT). Quality of life was evaluated using the Parkinson’s Disease Questionnaire (PDQ-39).”

- 297, educational paper material, it would be good to clarify what this means.

Response: We clarified this information as requested (page 26, lines 332-336).

“…the study of Rios Romenets et al. (2015) [86] differed from the other RCTs because its control group associates pharmacological usual care with a self-administered learning intervention (i.e. participants recei

---

## [Decision Letter · Decision Letter 1]

11 Sep 2024

Quality assessment and umbrella review of systematic reviews about dance for people with Parkinson's disease

PONE-D-23-44080R1

Dear Dr. Camila Pinto; pleased to inform you that your manuscript has been judged scientifically suitable for publication and will be formally accepted for publication once it meets all outstanding technical requirements.

Within one week, you receive an e-mail detailing the required amendments. When these have been addressed, you’ll receive a formal acceptance letter and your manuscript will be scheduled for publication.

An invoice will be generated when your article is formally accepted. Please note, if your institution has a publishing partnership with PLOS and your article meets the relevant criteria, all or part of your publication costs will be covered. Please make sure your user information is up-to-date by logging into Editorial Manager at Editorial Manager and clicking the ‘Update My Information' link at the top of the page. If you have any questions relating to publication charges, please contact our Author Billing department directly at authorbilling@plos.org.

Kind regards,

Mansueto Gomes Neto, Ph.D

Academic Editor

PLOS ONE

Additional Editor Comments (optional):

Reviewers' comments:

Reviewer's Responses to Questions

**Comments to the Author**

1. If the authors have adequately addressed your comments raised in a previous round of review and you feel that this manuscript is now acceptable for publication, you may indicate that here to bypass the “Comments to the Author” section, enter your conflict of interest statement in the “Confidential to Editor” section, and submit your "Accept" recommendation.

Reviewer #1: All comments have been addressed

2. Is the manuscript technically sound, and do the data support the conclusions?

Reviewer #1: Yes

3. Has the statistical analysis been performed appropriately and rigorously? 

Reviewer #1: I Don't Know

4. Have the authors made all data underlying the findings in their manuscript fully available?

Reviewer #1: Yes

5. Is the manuscript presented in an intelligible fashion and written in standard English?

Reviewer #1: Yes

6. Review Comments to the Author

Reviewer #1: The authors have done a thorough job of responding to both reviewers comments, I am satisfied that everything has been addressed and the paper is now suitable for publication.

7. PLOS authors have the option to publish the peer review history of their article (what does this mean?). If published, this will include your full peer review and any attached files.

Reviewer #1: **Yes: **Jennifer Hall

---

## [Editor Report · Acceptance letter]

25 Sep 2024

PONE-D-23-44080R1 

PLOS ONE

Dear Dr. Pinto, 

I'm pleased to inform you that your manuscript has been deemed suitable for publication in PLOS ONE. Congratulations! Your manuscript is now being handed over to our production team.

Kind regards, 

on behalf of

Dr. Mansueto Gomes Neto 

Academic Editor

PLOS ONE